# Redesigning regulatory components of quorum-sensing system for diverse metabolic control

Chang Ge[1], Zheng Yu[1], Huakang Sheng[1], Xiaolin Shen [1], Xinxiao Sun[1], Yifei Zhang [1], Yajun Yan [2], Jia Wang [1✉] & Qipeng Yuan [1✉]

Quorum sensing (QS) is a ubiquitous cell–cell communication mechanism that can be employed to autonomously and dynamically control metabolic fluxes. However, since the functions of genetic components in the circuits are not fully understood, the developed QS circuits are still less sophisticated for regulating multiple sets of genes or operons in metabolic engineering applications. Here, we discover the regulatory roles of a CRP-binding site and the lux box to −10 region within *luxR-luxI* intergenic sequence in controlling the lux-type QS promoters. By varying the numbers of the CRP-binding site and redesigning the lux box to −10 site sequence, we create a library of QS variants that possess both high dynamic ranges and low leakiness. These circuits are successfully applied to achieve diverse metabolic control in salicylic acid and 4-hydroxycoumarin biosynthetic pathways in *Escherichia coli*. This work expands the toolbox for dynamic control of multiple metabolic fluxes under complex metabolic background and presents paradigms to engineer metabolic pathways for high-level synthesis of target products.

[1] State Key Laboratory of Chemical Resource Engineering, Beijing University of Chemical Technology, Beijing 100029, China. [2] School of Chemical, Materials, and Biomedical Engineering, College of Engineering, The University of Georgia, Athens, GA 30602, USA. ✉email: wangjia@mail.buct.edu.cn; yuanqp@mail.buct.edu.cn

The employment of dynamic regulation circuits for precise balancing of metabolic fluxes between cell growth and metabolic production has emerged as a powerful strategy for economically producing target products[1]. Recent studies have focused on autonomous control of metabolic fluxes by designing genetic elements that can sense intracellular metabolites, such as acyl-CoA[2], malonyl-CoA[3], and pyruvate[4]. However, this approach requires pathway-specific biosensors and thus is inapplicable for the pathways without such biosensors. Quorum sensing (QS), a mechanism for cell–cell communication, is responsible for regulating cell density-dependent behaviors in bacteria[5]. Once the concentration of signaling molecules reaches a certain threshold, it will bind to a regulatory protein to trigger the expression of target genes under the control of the QS promoter[6]. Thus, the QS system represents a pathway-independent tool for autonomously regulating metabolic flux in response to cell density[7].

Nevertheless, the employment of natural QS circuits for the regulation of metabolic fluxes at different cell densities or expression levels is challenging as they are evolved to tightly control the expression of target genes within a narrow dynamic range[8]. Previous pioneering studies expanded the dynamic range of the lux-type QS system by applying pre-characterized promoters or ribosome binding sites with different strengths to drive the expression of partial QS components[9–11]. However, in most cases, these QS circuits can control the expression of one set of genes because only one type of QS system could be functional in a single cell, limiting their application in more challenging synthetic pathways. To address this issue, Prather and coworkers combined Lux-type and Esa-type QS systems to simultaneously up- and downregulate the expression of two different sets of genes in one cell[12]. However, the crosstalk between two QS systems could be a limit that prevents precisely controlling individual metabolic fluxes[13]. Although a peptide pheromone responsive QS system was identified to exhibit no crosstalk with lux-type QS system[14], the toxic peptide responsive QS system brings a metabolic burden to the host cells. Besides, the expression leakage of natural QS systems would also hinder cell growth when producing toxic proteins or chemicals. Therefore, it is highly desired to develop more sophisticated QS circuits with high dynamic ranges, low leakiness, and the ability to simultaneously regulate multiple sets of genes in one cell.

As one of the most widely used QS systems, the lux-type one from *V. fischeri* consists of three components, a transcriptional regulator protein LuxR, a signaling molecule acyl-homoserine lactones (AHLs) synthase LuxI, and a *luxR-luxI* intergenic sequence containing promoters $P_{luxR}$ and $P_{luxI}$ (Fig. 1a)[15,16]. The *luxR-luxI* intergenic sequence contributes to regulating the transcription of $P_{luxI}$ by binding with LuxR-AHL complex and RNA polymerase (RNAP)[17,18]. It plays a significant regulatory role in controlling the luminescence output of *V. fischeri*. Previous studies mainly focused on the mutational effects of the "lux box" sequence[19,20]. However, simple mutation of this sequences is not helpful to widen the dynamic ranges of QS circuits, and in some cases even leads to higher leakiness or the loss of responsiveness to signaling molecules[20]. In order to rationally engineer the functions of QS systems, it is necessary to reveal the regulatory mechanism of the intergenic sequence and identify novel target sites.

In this study, we elucidate the regulatory roles of the CRP-binding site and the lux box to −10 region within *luxR-luxI* intergenic sequence in controlling the lux-type QS promoters. We find that increasing the number of CRP-binding sites can significantly enhance the dynamic ranges and reduce leaky expression of the promoter $P_{luxI}$. By varying the numbers of CRP-binding sites and reconfiguring the lux box to −10 region sequence, we establish a library of 40 QS circuit variants that can control gene(s) expression in response to different concentrations of signaling molecules. As a characterization, these QS variants are able to autonomously and temporally regulate three different reporter genes in a single cell. We further confirm that integration of the CRISPR interference (CRISPRi) system into the lux-type QS system can switch the upregulation function into the downregulation function. Finally, we employ our designed QS circuits in the biosynthesis of salicylic acid (SA). By simultaneously regulating three different intracellular metabolic fluxes, we achieve the currently highest SA titer of 2.08 g/L in shake flasks. We implement these circuits for dynamic modular optimization of 4-hydroxycoumarin (4-HC) production, yielding a 10-fold improvement in titer compared to that of the strain without dynamic flux control. These results suggest the QS circuits possess broad applications in the field of metabolic engineering.

## Results

**Investigating the regulation mechanism of *luxR-luxI* intergenic sequence.** Autonomous and temporal control of multiple metabolic fluxes requires developing a series of QS promoters that can trigger the expression of each set of genes at a different time in a single cell. Such QS circuits could be created by in situ engineering of the natural QS system, which, however, are challenging due to the unknown regulation mechanism. In the natural QS system, the *luxR-luxI* intergenic sequence is responsible for binding with the LuxR-AHL complex and RNAP to initiate the transcription of promoter $P_{luxI}$ (Fig. 1a)[21]. We hypothesized that mutation of this region is able to create a variety of QS variants that have a different binding affinity towards the LuxR-AHL complex and RNAP. When co-expressing these circuits in one cell, they will exhibit different QS behaviors. Random mutation of the *luxR-luxI* intergenic sequence has been proved to be ineffective in generating well-performed QS circuits[20]. Rational engineering of this sequence requires a deep understanding of the functions of genetic components, which has yet been clearly elucidated so far.

It has been reported that there is a CRP-binding site located in the *luxR-luxI* intergenic sequence (Fig. 1a)[22]. The CRP-binding site is an important transcriptional regulation sequence (Consensus sequence: AAATGTGATCTAGATCACATTT) widely distributed in many bacteria[23,24]. The CRP is an allosteric protein that usually binds to its signaling molecule cyclic AMP to form dimers or tetramers. This CRP-cAMP complex can bind to a recognition site in the target core promoters to assist RNA polymerase binding to the promoter[25]. However, the exact function of the CRP-binding site in the lux-type QS system has not yet been proposed so far. Considering that the location of the CRP-binding site is very close to $P_{luxR}$, we hypothesized that $P_{luxR}$ is a CRP-dependent promoter. The increased numbers of the CRP-binding site could recruit more CRP-cAMP complex to enhance the transcription of $P_{luxR}$ by strengthening the interactions with RNAP. To test this hypothesis, we arranged in tandem multiple copies of the CRP-binding site ($n = 0, 1, 2, 3, 4$) in the *luxR-luxI* intergenic sequence (Supplementary Fig. 1a). The reporter gene *mCherry* was placed downstream of the promoter $P_{luxR}$. As we expected, with the increase of the CRP-binding site numbers, the transcriptional level of the gene under the control of the $P_{luxR}$ was dramatically enhanced (Fig. 1b and Supplementary Fig. 2). The strain-carrying QS circuits with four copies of the CRP-binding site produced eightfold higher peak fluorescence than the strain harboring QS circuits without a CRP-binding site, indicating that the presence of more CRP-binding sites is beneficial for $P_{luxR}$ transcription. In order to confirm how many CRP-binding sites are best, we increased the CRP-binding sites to

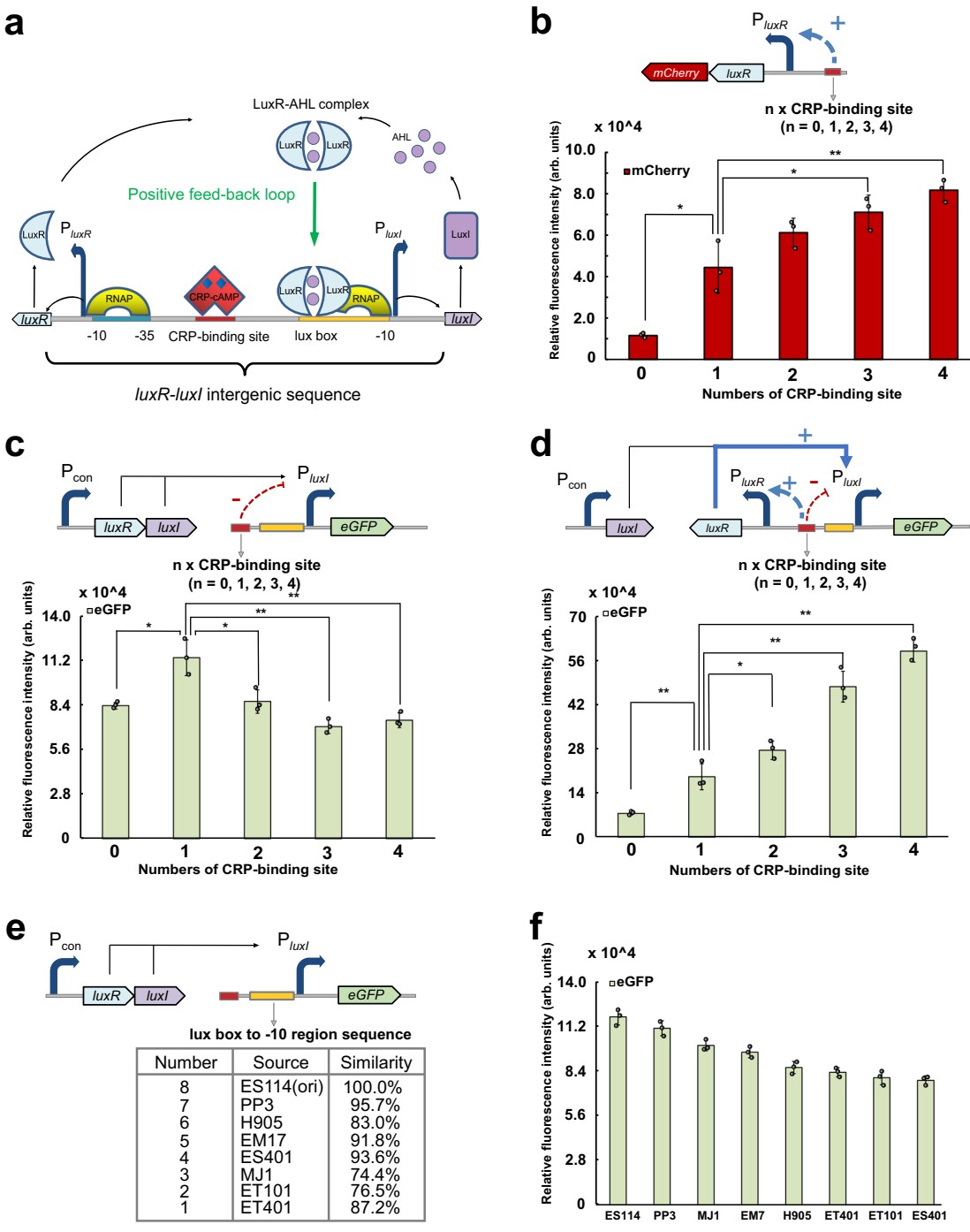

**Fig. 1 Investigation of the regulatory mechanism of *luxR-luxI* intergenic sequence. a** Schematic diagram of the Lux-type QS system. **b** The strength of $P_{luxR}$ promoter with tandem copies of CRP-binding site ($n = 0, 1, 2, 3, 4$). *P* values: 0.01, 0.11, 0.04, and 0.01. **c** The strength of $P_{luxI}$ promoter with tandem copies of CRP-binding site ($n = 0, 1, 2, 3, 4$). The genes *luxR* and *luxI* were expressed under the control of a constitutive promoter. *P* values: 0.01, 0.02, 0.03, and 0.05. **d** The strength of $P_{luxI}$ promoter with tandem copies of CRP-binding site ($n = 0, 1, 2, 3, 4$) in the presence of $P_{luxR}$ controlled LuxR. The gene *luxI* was expressed under the control of a constitutive promoter. *P* values: 0.01, 0.04, 0.00, and 0.00. **e** The sources of the "lux box" to "−10" site sequence and their sequence similarity with ES114. **f** Fluorescence intensity of each $P_{luxI}$ promoter with different "lux box" to "−10" site sequence. The genes *luxR* and *luxI* were expressed under the control of a constitutive promoter. AHL acyl-homoserine lactone, LuxI AHLs synthase, LuxR a transcriptional regulator protein, cAMP cyclic adenosine monophosphate, CRP-cAMP receptor protein, CRP-cAMP a complex of CRP binding to cAMP, RNAP RNA polymerase. Data shown are mean ± SD ($n = 3$ independent experiments). The statistical analysis is based on a two-sided Student's *t*-test. *$P < 0.05$ and **$P < 0.025$. Source data are provided as a Source Data file.

7. As shown in Supplementary Fig. 3, when the CRP-binding site varies from 0 to 7, the QS promoter harboring four CRP-binding sites exhibited the highest transcription level and dynamic range, indicating that four CRP-binding sites is the best. We further performed electrophoretic mobility shift assay (EMSA) experiments to confirm the result. As shown in Supplementary Fig. 4a, b, as the number of CRP-binding sites increases, the number of CRP proteins bound to the DNA increases too, indicating that more CRP-binding sites promoted the CRP protein binding to the DNA. We also used an atomic force microscope to observe the topographies of the CRP-DNA complex. As shown in Supplementary Fig. 5, when incorporating four CRP-binding sites in the locus, the DNA and CRP protein forms obvious multimers compared with that of one CRP-binding site in the locus. We believe that the formed multimers might promote the bounded CRP to recruit more RNAP for transcription. Since $P_{luxR}$ is a type of CRP-regulated promoter, it is necessary to further explore which class it belongs to. The amino acid H159 in the CRP protein was substituted for Leu and the amino acid V287 in the alpha subunit of RNAP was substituted for Ala (Supplementary Fig. 6a, b). The results showed that both substitutions dramatically reduced the transcription of $P_{luxR}$ (Supplementary Fig. 6c), indicating that $P_{luxR}$ appears to be a class I CRP-dependent promoter.

Then, we began to investigate how the CRP-binding sites would control the transcription of $P_{luxI}$. The $luxR$ and $luxI$ genes were expressed under the control of a constitutive promoter to make sure the number of LuxR or LuxI is the same in all strains. The reporter gene $eGFP$ was placed downstream of the promoter $P_{luxI}$. As shown in Fig. 1c, complete deletion of the CRP-binding site in the $luxR$-$luxI$ intergenic sequence decreased the transcriptional intensity of $P_{luxI}$. The existence of one copy CRP-binding site ($n = 1$) promotes the transcription of $P_{luxI}$. However, further increase of the CRP-binding site numbers ($n = 2, 3, 4$) inhibited the gene expression under the control of the promoter $P_{luxI}$. We further performed EMSA experiments to confirm this result. As shown in Supplementary Fig. 7 and 8, the CRP protein indeed assists RNAP binding to the promoter $P_{luxR}$. When incorporating four CRP-binding sites in the locus, the binding band of RNAP with $P_{luxR}$ showed no significant difference compared with that of one CRP-binding site in the locus. However, the binding band of LuxR and $P_{luxI}$ was obviously shallowed. Based on these results, we believe that a high local concentration of CRP-cAMP might sterically hinder the binding of the LuxR-AHL complex to the promoter $P_{luxI}$.

We have demonstrated that an increase of CRP-binding site numbers activates the transcription of the $P_{luxR}$ while inhibiting the transcription of $P_{luxI}$. It is worth noting that the activated $P_{luxR}$ would further enhance the expression level of LuxR, leading to strong activation of the transcription of $P_{luxI}$[9]. When both $P_{luxR}$ and $P_{luxI}$ exist in the QS system, how the increased CRP-binding site numbers would control the transcription of promoter $P_{luxI}$ should be further investigated. Thus, we placed LuxR under the control of the $P_{luxR}$ promoter, and the expression of reporter gene $eGFP$ was driven by the $P_{luxI}$ promoter. We observed that the expression level of $eGFP$ was increased by increasing the CRP-binding site numbers (Fig. 1d). The QS system with four copies of the CRP-binding site ($n = 4$) exhibited the highest $P_{luxI}$ activity, which is 12.1-fold higher than that of the QS system with no CRP-binding site ($n = 0$). This result indicates that an increase of the CRP-binding site numbers could significantly expand the dynamic range of $P_{luxI}$ when co-expressing the $P_{luxR}$ controlled $luxR$ in the system and $luxI$ expressed from a constitutive promoter. We found that integration of more CRP-binding sites in the $luxR$-$luxI$ intergenic sequence inhibits the transcription of $P_{luxI}$ (in the absence of $P_{luxR}$) but significantly enhances the

transcription of $P_{luxR}$ (in the absence of $P_{luxI}$). However, when both $P_{luxI}$ and $P_{luxR}$ controlled $luxR$ co-exist in the QS system, the activated $P_{luxR}$ strongly activates $P_{luxI}$ transcription. In this condition, an increase in the CRP-binding site numbers greatly improves the transcription level of $P_{luxI}$.

Next, we focused on identifying other target sites in the $luxR$-$luxI$ intergenic sequence. It has been reported that the luminescence output varies over orders of magnitude between different $Vibrio$ species[26]. By sequencing and aligning the genomes of these diverse $Vibrio$ strains, significant sequence differences were found in the lux box to the $-10$ site of the $P_{luxI}$ promoter. Based on this, we hypothesized that the binding affinity could be altered by mutation of the lux box to $-10$ site sequence via directly using the homologous sequence from these different $Vibrio$ strains. Thus, we used the natural QS system from $V.$ $fischeri$ ES114[27] as a template and created seven QS variants by replacing its lux box to $-10$ site sequence with the homologous one from other seven $Vibrio$ species, including PP3[28], ES401[29], ET401[30], EM17[29], ET101[31], H905[28], and MJ1[19] (Fig. 1e). The lux box to $-10$ site sequence alignment showed that they shared high similarity (74.5%~95.7%) with $V.$ $fischeri$ ES114 (Supplementary Fig. 1b). To characterize circuit behaviors, we constructed eight pZE plasmids by individually employing these eight QS systems to drive green fluorescent protein (GFP) expression. The plasmid pCS-$P_{con}$-LuxR-LuxI was constructed for constitutively expressing genes $luxR$ and $luxI$. It was co-transferred with each pZE plasmid into $E.$ $coli$ BW25113 (F'), yielding eight fluorescent strains. As we expected, these strains generated a variety of fluorescent profiles by continuous fluorescence measurements (Fig. 1f). Generally, the use of the homologous sequence to replace the natural one slightly decreased the transcription of $P_{luxI}$, the strain harboring the lux box to $-10$ site sequence from $Vibrio$ ES401 exhibited the lowest peak fluorescence, which is 2-fold lower than that of the $Vibrio$ ES114 (Fig. 1f). The EMSA experiments were carried out to test the relative binding strength of QS systems harboring different sources of lux box to $-10$ site sequence (Supplementary Fig. 9a). The results showed that the use of the homologous sequence to replace the natural one (ES114) decreased the relative binding strength of the QS system. The trends were well-matched with their transcriptional intensities (Supplementary Fig. 9b). These results indicate that mutation of the lux box to the $-10$ site region in the $luxR$-$luxI$ intergenic sequence has only a slight inhibition effect on the transcriptional intensity of promoter $P_{luxI}$. However, we believe that a couple of this mutation with CRP-binding site number variation could further expand the diversity of QS variants. probably these mutations reduced the binding affinity between the promoter sequence and the transcriptional factor complex.

**Establishing and characterizing a library of QS variants.** We have created a series of QS variants that have different behaviors by mutation of the lux box to $-10$ site sequence and variation of the CRP-binding site numbers. Then, these two strategies were combined to construct a library of QS promoters with different triggering times (Fig. 2a). By altering the numbers of the CRP-binding site ($n = 0, 1, 2, 3, 4$) and the sequence of the lux box to the $-10$ site region (ES114(8), PP3(7), H905(6), EM17(5), ES401(4), MJ1(3), ET101(2), and ET401(1)), a total of 40 QS variants were generated. In order to obtain high dynamic range QS promoters, the $P_{luxR}$ controlled $luxR$ was also integrated into these systems. Continuous fluorescence measurements of these variants were performed to characterize their performance. As shown in Figs. 2b–d, these 40 QS variants exhibited a wide range of peak fluorescence intensities and triggering times. Generally, as the numbers of the CRP-binding site increased, peak GFP

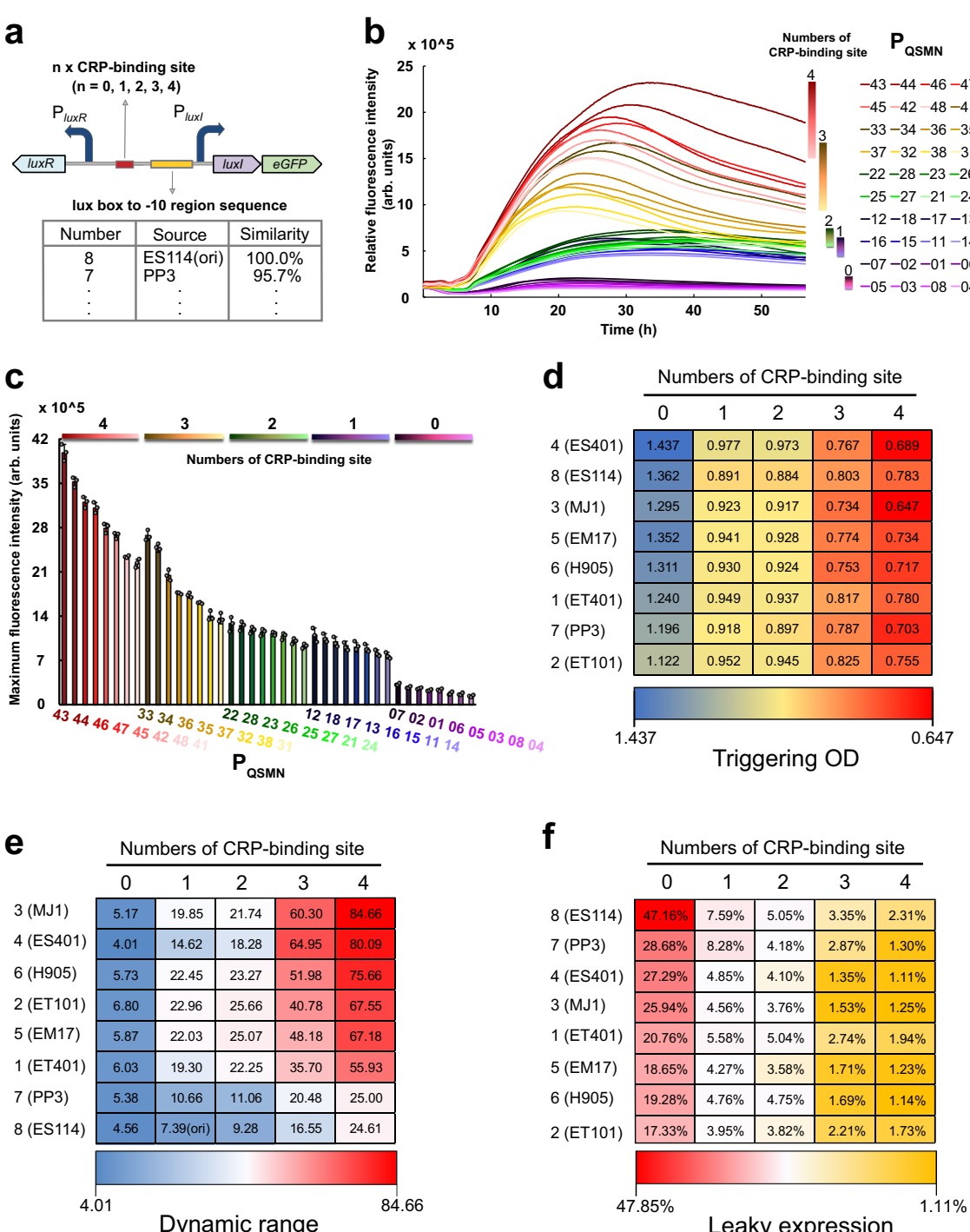

**Fig. 2 Establishment and characterization of QS variant library. a** Architecture of the QS variant in the library. **b** Gene expression dynamics of each QS variant in the library. Each QS variant was named $P_{QSMN}$, "M" represents the copy number of the CRP-binding site carried by the QS variant. "N" represents the source of the "lux box" to "−10" site sequence contained by the QS variant. **c** The peak fluorescence intensity of each QS variant in the library. **d** Triggering OD for each QS variant in the library. **e** Dynamic range for each QS variant. **f** Leaky expression for each QS variant. The leaky expression was exhibited as the ratio of comparing peak fluorescence intensity from strains carrying partial QS circuits in the absence of *luxR* and *luxI* to that of strains harboring entire QS circuits in the presence of *luxR* and *luxI*. Data shown are mean ± SD (*n* = 3 independent experiments). Source data are provided as a Source Data file.

fluorescence increased too (Fig. 2b). The strain carrying the QS circuit with no CRP-binding site and the lux box to −10 site sequence from *Vibrio* ES401 produced the lowest peak fluorescence. The highest peak fluorescence was observed in the strain harboring the QS circuit with 4 CRP-binding sites and the lux box

to −10 site sequence from *Vibrio* MJ1, which is 28.3-fold higher than that of the lowest strain (Fig. 2c). To compare the triggering times of these QS variants, we used the peak fluorescence value of the QS variant that has the lowest fluorescence signal in the library as a reference value. The "triggering OD" was defined as

the cell density corresponding to the time of fluorescence reaching the reference value (Supplementary Fig. 10). As shown in Fig. 2d, triggering ODs of these variants ranges from 0.647 to 1.437, which covers the growth period from the early logarithmic phase to the stationary phase. We found that the triggering OD was decreased by increasing the number of CRP-binding sites. We tested the AHL accumulation rates of the QS systems harboring different CRP-binding sites ($n = 0, 1, 2, 3, 4$). As shown in Supplementary Fig. 11, the AHL accumulation rate was increased by increasing the number of CRP-binding sites, indicating that the decreased triggering OD resulted from the increased AHL accumulation rates. Altering the lux box to the −10 site sequence has only a slight effect on the triggering ODs under the same CRP-binding site number. These trends are consistent with our expectations based on the regulatory mechanism discovered in this study.

To validate the function of these QS variants, we evaluated their dynamic range and leakiness. The dynamic range is defined as the ratio of the peak GFP fluorescence value of the strain carrying entire QS elements to that of the strain harboring partial QS circuits lacking the *luxI* gene. As seen in Fig. 2e and Supplementary Fig. 12a, we found that these QS variants exhibited a broad spectrum of dynamic ranges, ranging from 4.0-fold to 84.7-fold. It is worth noting that the highest dynamic range observed in the QS variant is 11.5-fold higher than the natural QS system without any modifications. For most promoters, improvement of their dynamic range usually leads to enhanced leakiness. Thus, we also evaluated the expression leakage of these QS variants, which was calculated as the ratio of comparing peak fluorescence intensity from the strains carrying partial QS circuits in the absence of *luxR* and *luxI* to the strains harboring entire QS circuits in the presence of *luxR* and *luxI*. Interestingly, except for QS variants lacking the CRP-binding site ($n = 0$), almost all of the QS variants in the library exhibited a declined leaky expression compared with the natural QS system from *V. fischeri* ES114 (Fig. 2f and Supplementary Fig. 12b). The leakiness of the QS variant was decreased by increasing the CRP-binding site numbers. Use of the homologous lux box to −10 site region sequences to replace the natural one also slightly reduced leaky expression. Especially, the QS variant harboring 4 CRP-binding sites and lux box to −10 site sequence from ES401 exhibited 6.7-fold lower leakiness than the natural QS system. These results indicate that variation of the lux box to −10 site sequence and increase of the CRP-binding site numbers reduced the leaky expression of QS systems, due to these mutations inhibited the transcription of P$_{luxI}$ promoter. These finely tuned QS variants pave the way for implementing dynamic controls in complex metabolic networks.

**Application of QS variants for autonomous and temporal regulation of multiple gene targets**. Since our constructed QS variants have a high dynamic range, low leakiness, and different triggering times, those circuits can be used to simultaneously regulate three or more metabolic fluxes in a single cell, which has not yet been achieved before. We hypothesized that sequential transcription of multiple sets of genes can be achieved by regulating their expression under the control of the different QS variants. To test our hypothesis, we selected three QS variants with different triggering OD as 0.79, 0.89, and 0.92 to control the expression of three different reporter genes *egfp*, *mCherry*, and *mOrange*2, respectively. We constructed the plasmid pZE-*luxR*-P$_{QS37}$-*luxI-eGFP*-P$_{QS18}$-*mCherry*-P$_{QS13}$-*mOrange2* and transferred it into *E. coli*, yielding strain CGemm (Fig. 3a). We measured the fluorescence of strains that carry no reporter proteins as negative controls (Supplementary Fig. 13). The normalized

relative fluorescence intensities of these three reporter proteins in the strain CGemm were measured and compared (Fig. 3b). As we expected, the green fluorescent protein eGFP under the control of the QS variant P$_{QS37}$ was expressed first. Then, the expression of the red protein mCherry was initiated about one hour later. Finally, the yellow fluorescence produced by the protein mOrange2 under control of the QS variant P$_{QS13}$ was detected. The expression order of these three reporter genes observed through fluorescence characterization matches well with the predicted order. To further confirm the consistency, we used a laser scanning confocal microscope to directly observe the fluorescence of the strain CGemm. As shown in Fig. 3c, the green, red, and yellow fluorescence was sequentially observed in the fluorescent images, which is consistent with the predicted expression order. In order to confirm the sequential expression is not caused by the different maturation times of the reporter proteins, we placed the *eGFP* under the control of the QS promoters P$_{QS37}$, P$_{QS13}$, and P$_{QS18}$ by constructing plasmids P$_{QS37}$-*eGFP*-CO, P$_{QS13}$-*eGFP*-CO, and P$_{QS18}$-*eGFP*-CO, respectively. We tested the fluorescence profiles of the strains harboring these circuits. As shown in the Supplementary Fig. 14, when using P$_{QS37}$, P$_{QS13}$, and P$_{QS18}$ to drive the expression of the same reporter gene, the expression sequence is consistent with that in the Fig. 3b, c, indicating that the sequential expression is not caused by the different maturation time of the reporter proteins. These results indicate that our constructed QS variants are reliable tools to temporally and autonomously control the expression of multiple gene targets in a single cell.

The above-established QS circuits only have the function of activating transcription. To expand the functionality of the QS system, we tested the potential of using CRISPRi as a switch to achieve signal conversion. Thus, the AHL-induced upregulation was transduced into a downregulation function by integrating the CRISPRi system into QS circuits. To confirm whether simultaneously dynamic upregulation and downregulation can be achieved in an orthogonal manner, we constructed plasmid pZE-*luxR*-P$_{QS18}$-*luxI-eGFP* as the reporter of the upregulation function. To construct the downregulation function reporter, the plasmid plv-P$_{QS18}$-*dCas9-sgRNA* was constructed to repress the *mCherry* expression on the genome of *E. coli* BW25113 (F′) (Fig. 3d). As seen from Fig. 3e, in the presence of the AHL and CRISPRi system, the eGFP fluorescence intensity was upregulated by 7.7-fold, and the mCherry fluorescence intensity was downregulated by 82.9% by the end of 24 h. This result demonstrates that coupling the CRISPRi system with the QS variants could generate a dynamic and bi-functional regulation tool for simultaneously up- and downregulate two sets of genes.

**Dynamic regulation of salicylic acid biosynthesis**. Next, we chose the biosynthesis of SA as a model to verify the effectiveness of designed genetic circuits in regulating intracellular metabolic fluxes. SA is a value-added chemical that has been used to produce medicines, perfumes, and dyes. In *E. coli*, biosynthesis of SA can be achieved by extending the shikimate pathway via overexpression of *entC* and *pchB* (Fig. 4a)[32]. However, SA is toxic to the *E. coli* strain and 1 g/L SA completely inhibits the cell growth[32]. We hypothesized that our designed QS variants could be used to improve SA titer by switching on the expression of pathway genes at appropriate time points. To test this hypothesis, QS promoters P$_{QS13}$, P$_{QS18}$, P$_{QS37}$, and P$_{QS33}$ from the library (Fig. 2b) were individually employed for driving the expression of *entC* and *pchB*, generating four SA upregulation modules. As a control, *entC* and *pchB* were constitutively expressed to provide a statically controlled SA upregulation module. As shown in Fig. 4b, the static expression strain only generated 295.1 mg/L SA.

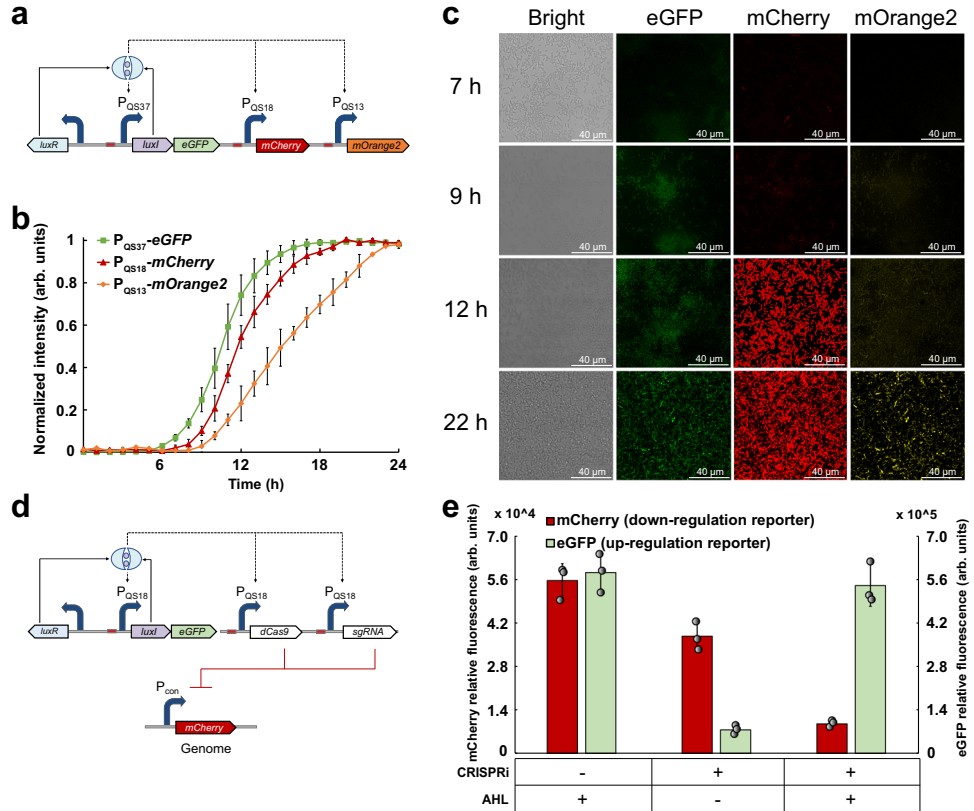

**Fig. 3 Autonomous and simultaneous control of three different gene targets by QS variants. a** Architecture of the QS circuit that simultaneously regulates the expression of *eGFP*, *mCherry*, and *mOrange*2, under control of the P<sub>QS37</sub>, P<sub>QS18</sub>, and P<sub>QS13</sub>, respectively. **b** Fluorescence profiles for *eGFP*, *mCherry*, and *mOrange*2 in the strain containing the designed QS circuit. For all three fluorescent proteins, the fluorescent intensity of each time point was normalized by their corresponding peak fluorescence value. **c** Laser scanning confocal images captured in a single cell containing the designed QS circuit. Images were taken at 7, 9, 12, and 22 h and under three different excitation wavelengths. Bar = 40 μm. The data were generated from three independent experiments. **d** Architecture of the QS circuit for simultaneous up- and downregulation of two different gene targets. **e** Peak fluorescent intensities of mCherry and eGFP when introducing CRISPRi into this system. Data shown are mean ± SD (*n* = 3 independent experiments). Source data are provided as a Source Data file.

While, the strain harboring QS P<sub>QS13</sub> controlled pathway genes produced the highest SA titer of 523.2 mg/L, demonstrating a 72% increase compared with the static strain. It was also found that the strain carrying the QS variant with lower triggering OD produced less amount of SA. This demonstrates that delaying the expression of SA pathway genes is beneficial for improving the SA titer, probably due to the released metabolic burden to the host cell. Additionally, we observed no improvement in the SA titer in the absence of the *luxI*, indicating that the increase in SA production is induced by the QS dynamic regulation. We also tested the cell growth rates, glucose levels, and the SA titer of the SA-producing strains at different times (Supplementary Fig. 15a–c), the results showed that these factors make no contribution to the titer improvements.

Then, we aimed to precisely control SA production by using the QS variants to simultaneously regulate two metabolic fluxes. Having a sufficient supply of phosphoenolpyruvate (PEP) has been proved to be a key factor for the biosynthesis of the shikimate pathway-derived compounds[33]. Except as a precursor for the shikimate pathway, PEP also serves as an essential metabolite in glycolysis and the TCA cycle[34]. However, direct disruption of these competing pathways would impair cell growth[35–37]. Especially, direct deletion of the phosphoenolpyruvate carboxylase (encoded by *ppc*) in *E. coli* significantly reduced cell growth[35]. Alternatively, we attempted to dynamically down-regulate *ppc* to balance the relationship between production and growth. The QS promoter P<sub>QS18</sub> was used to control the expression of the CRISPRi system. We designed four sgRNAs targeting different locations of *ppc* operon, including ribosome binding site (*sgppc*1), start codon region (*sgppc*2), coding sequence at 261-bp (*sgppc*3), and coding sequence at 1406-bp (*sgppc*4). This generated four different downregulation modules that can inhibit the expression of *ppc* at different levels (*sgppc*1 > *sgppc*2 > *sgppc*3 > *sgppc*4) (Supplementary Fig. 3a). Additionally, as a control, we constructed the fifth down-regulation module by complete disruption of *ppc*. These five downregulation modules were combined with the former constructed five SA upregulation modules, yielding a library of 25 SA-producing strains. Generally, complete deletion or high-level downregulation (*sgppc*1 and *sgppc*2) of *ppc* resulted in only negligible SA titers, probably due to the cell growth retardation (Fig. 4c and Supplementary Fig. 16b). While higher SA titers were observed in dynamic downregulation of *ppc* at lower levels using *sgppc*3 and *sgppc*4. By quantitative real-time PCR analysis, the use of *sgppc*3 and *sgppc*4 inhibited the expression level of *ppc* by 42.4% and 33.6%, respectively (Supplementary Fig. 16a). Among the dual dynamically regulated strains that contain both upregulation module and downregulation module, downregulating the expression of *ppc* using *sgppc*4 and placing the SA pathway genes under the control of the P<sub>QS37</sub> is the optimal combination for SA production. This optimal SA producer generated 1010.6 mg/L SA after 48 h fermentation. As a control,

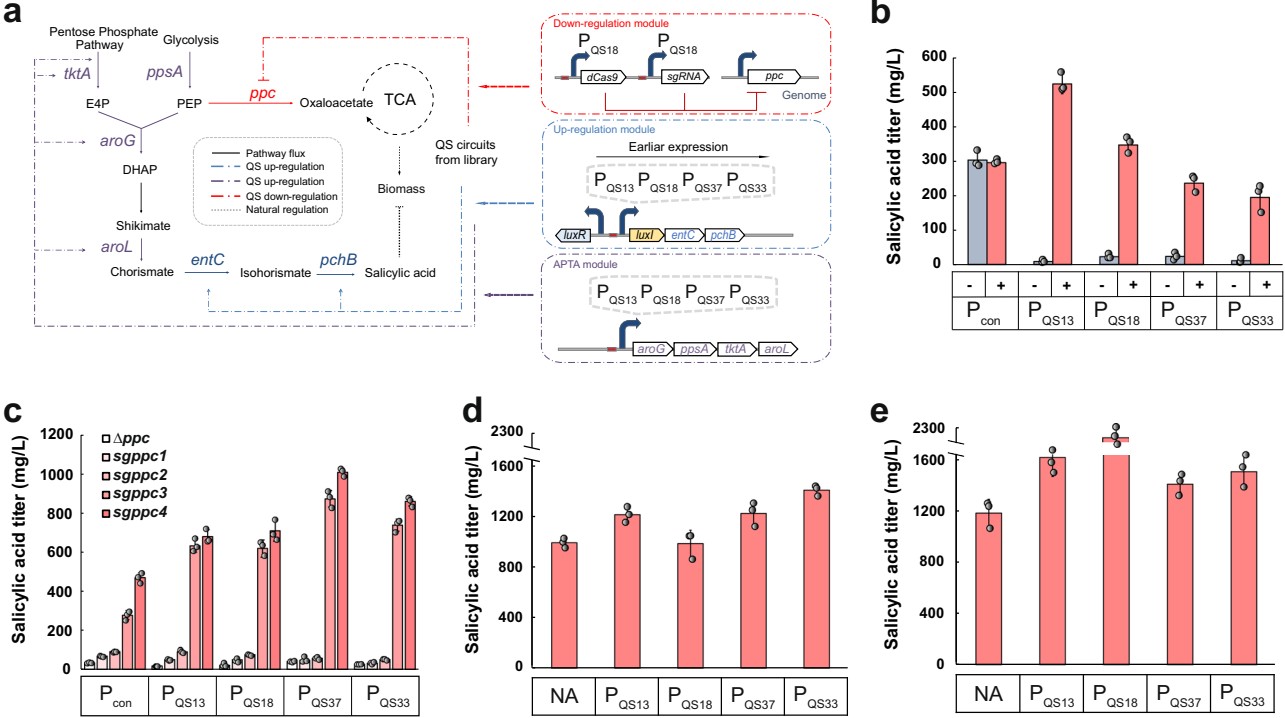

**Fig. 4 Enhancement of SA production by dynamically and simultaneously controlling multiple metabolic fluxes. a** Schematic construction of layered dynamic control circuits for regulation of the SA biosynthetic pathway. **b** The effect of different triggering OD of QS variants driving *entC-pchB* expression on SA titer. Gray bars represent SA titers generated by the strains harboring partial QS circuits in the absence of the gene *luxI*. Red bars represent SA titers generated by the strains harboring entire QS circuits in the presence of the gene *luxI*. **c** The effect of different triggering OD of QS variants driving *entC-pchB* combined with different *ppc* downregulation levels on SA titer. **d** The effect of three-layered dynamic regulation strategy on SA titer in wildtype *E. coli* strain. The upregulation module containing SA pathway genes was controlled by the QS variant $P_{QS37}$, the downregulation module carrying the CRISPRi system was controlled by the QS variant $P_{QS18}$, and the APTA module was controlled by QS variant $P_{QS13}$, $P_{QS18}$, $P_{QS37}$, or $P_{QS33}$, respectively. **e** The effect of three-layered dynamic regulation strategy on SA titer in QH4 strain. The upregulation module containing SA pathway genes was controlled by the QS variant $P_{QS37}$, the downregulation module carrying the CRISPRi system was controlled by the QS variant $P_{QS18}$, the APTA module was controlled by QS variant $P_{QS13}$, $P_{QS18}$, $P_{QS37}$, or $P_{QS33}$, respectively. NA represents the SA titer generated by the control strain lacking the APTA module. Genes: *tktA* encodes transketolase, *ppsA* encodes phosphoenolpyruvate synthase, *aroG* encodes 3-deoxy-D-arabinoheptulosonate-7-phosphate (DHAP) synthase, *aroL* encodes shikimate kinase, *ppc* encodes phosphoenolpyruvate carboxylase, *entC* encodes isochorismate synthase, *pchB* encodes isochorismate pyruvate lyase. Metabolites: E4P *D*-erythrose-4-phosphate, PEP phosphoenolpyruvate. Data shown are mean ± SD ($n = 3$ independent experiments). Source data are provided as a Source Data file.

SA biosynthesis was conducted by static overexpression of SA pathway genes and downregulation of *ppc* using *sgppc*4. As another control, SA pathway genes were placed downstream of the $P_{QS37}$, and this circuit was introduced into a host strain with *ppc* disruption. These two control strains only produced 470.8 and 35.5 mg/L SA, which were 2.1-fold and 28.5-fold lower than that of the optimal SA producer. These results demonstrate that the timing of target pathway expression and competing pathway repression must be carefully tuned to identify the optimal SA producer. We also tested the final titer relative to the total substrate consumed, the final titer relative to the biomass, and SA titer dynamics over cultivation time (Supplementary Fig. 15d–f), the results showed that all of the trends are consistent with the final titers observed in the Fig. 4c.

As the next step, we examined the effect of simultaneous dynamic control of three metabolic fluxes on SA biosynthesis. We previously constructed a chorismate-boosting plasmid pCS-APTA that carries *aroL*, *ppsA*, *tktA*, and *aroG*<sup>fbr</sup> (a feedback inhibition resistant mutant of *aroG*)[38]. The QS promoters $P_{QS13}$, $P_{QS18}$, $P_{QS37}$, and $P_{QS33}$ were individually used to drive the expression of the APTA module, generating plasmids pSA-$P_{QS13}$-APTA, pSA-$P_{QS18}$-APTA, pSA-$P_{QS37}$-APTA, and pSA-$P_{QS33}$-APTA. These four plasmids were individually transferred into the above-obtained optimal SA producer, yielding four SA

producers that harbor QS circuits controlling three metabolic fluxes, as (1) the upregulation module containing SA pathway genes was controlled by the QS variant $P_{QS37}$, (2) the downregulation module carrying CRISPRi system was controlled by the QS variant $P_{QS18}$, and (3) the APTA module was controlled by QS variant $P_{QS13}$, $P_{QS18}$, $P_{QS37}$, or $P_{QS33}$, respectively. As seen from Fig. 4d, the addition of the third layer over-expresses APTA further enhancing the SA titers. Especially, the use of the $P_{QS33}$ for driving APTA module expression led to 1409.8 mg/L SA, showing a 43% increase compared with the strain that only has two layers of dynamic flux control. This indicates that the place of APTA downstream of the $P_{QS33}$ provided the optimal time for initiating the expression of the APTA module. Additionally, we tested the three-layered dynamic control circuits in a phenylalanine over-producer QH4, which is derived from the disruption of *pheA* and *tyrA* in *E. coli* ATCC31884. As shown in Fig. 4e, transform of the strain background from wildtype *E. coli* to QH4 led to a further increase in SA titers. The best-performed producer generated 2.1 g/L SA (546.5 mg/g CDW, 700 mg/L/day), which is the highest production achieved in shake flask experiments. Interestingly, we found that the optimal QS promoter for controlling the expression of APTA module in wildtype *E. coli* is $P_{QS33}$, with a triggering OD of 0.73. While, the highest SA titer in QH4 was observed when the APTA module

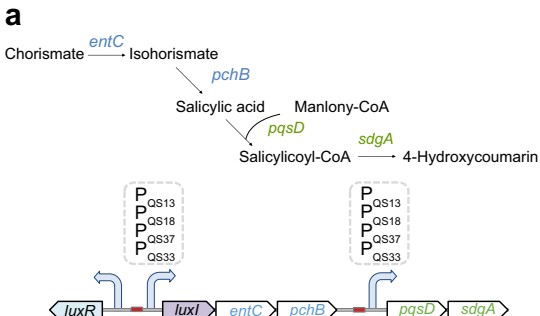

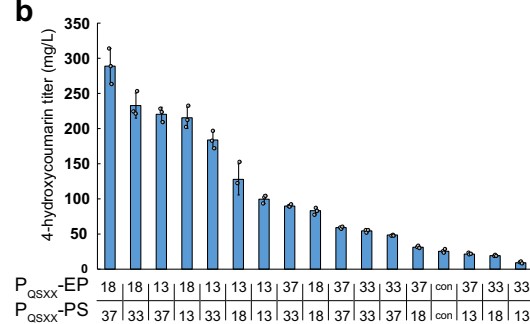

**Fig. 5 Application of QS variants for dynamic modular optimization of 4-HC production. a** Schematic of construction of dynamic control circuits for regulation of 4-HC biosynthetic pathway. **b** The effect of different triggering OD of QS variants individually driving *entC-pchB* expression and *pqsD-sdgA* expression on 4-HC titer. Con represents 4-HC titer generated by the control strain expressing EP and PS modules on constitutive promoters. Gene: *entC* encodes isochorismate synthase, *pchB* encodes isochorismate pyruvate lyase, *pqsD* encodes CoA ligase, *sdgA* encodes biphenyl synthases. Data shown are mean ± SD ($n = 3$ independent experiments). Source data are provided as a Source Data file.

was controlled by the QS promoter $P_{QS18}$, with a higher triggering OD of 0.89. This indicates that delayed overexpression of the APTA module in QH4 is beneficial for SA production. Owing to the carbon flux through the shikimate pathway in QH4 is higher than in wildtype *E. coli* strain, too earlier overexpression of APTA module in QH4 might bring metabolic burden to the host. Taken together, these results demonstrate that our designed QS variants are powerful tools for improving microbial production by simultaneously and dynamically regulating multiple metabolic fluxes.

**Dynamic modular optimization of 4-hydroxycoumarin production**. Modular optimization, including tuning gene expression level by altering plasmid copy number and engineering promoter or ribosomal binding site (RBS) strength, has been proved to be an effective strategy for pathway balancing[39–41]. However, the currently developed modular optimization method only can modulate enzyme expression levels on a static mode, which is unable to respond to intracellular and environmental changes. We hypothesized that our designed QS circuits could be applied to dynamically optimize carbon flux distribution among different pathway modules to improve production. We utilize the 4-HC biosynthetic pathway as a demonstration to verify our hypothesis. 4-HC is a value-added compound used in the production of anticoagulant drugs, and also plays an important role in treating thromboembolic diseases[42,43]. The 4-HC biosynthetic pathway consists of two modules, (1) EP module containing EntC and PchB that converts chorismate to SA, (2) PS module carrying salicyl-CoA ligase (SdgA) and biphenyl synthase (PqsD) that is responsible for generating 4-HC via condensation of SA with malonyl-CoA (Fig. 5a)[38]. The incongruous expression of the EP and PS modules results in metabolic imbalance, which serves as a rate-limiting factor for 4-HC synthesis[44]. Additionally, chorismate and malonyl-CoA are two essential metabolites in *E. coli*[45], indicating the trade-off between 4-HC production and cell growth. We hypothesized that the use of QS variants for dynamically modular optimization of EP and PS modules would improve 4-HC titer by balancing the carbon flux and initiating the expression of each module at the optimal time.

To verify the effectiveness of dynamic modular optimization on 4-HC production, four QS promoters $P_{QS13}$, $P_{QS18}$, $P_{QS37}$, and $P_{QS33}$ with different triggering OD were selected. Each of these QS promoters was applied to drive the expression of the EP module and/or PS module, resulting in a library of sixteen 4-HC producers. These 4-HC producing strains are able to switch on the expression of EP and PS modules at different rates in a dynamic and autonomous mode. As shown in Fig. 5b, these

strains produced varying 4-HC titers. The best producer harboring EP module controlled by the $P_{QS18}$ and PS module controlled by the $P_{QS37}$, generated the highest 4-HC titer of 288.8 mg/L, representing an 11.3-fold increase over the static producer. Generally, the EP module driving by QS promoter with higher triggering OD and PS module driving by QS promoter with lower triggering OD led to higher 4-HC titers, indicating that 4-HC production could be improved by fine-tuning the timing of each module expression. These results demonstrate the generalizability and effectiveness of our designed QS circuits for dynamic pathway engineering.

## Discussion
The trade-off between cell growth and product synthesis is a big challenge in metabolic engineering[46,47]. Applying a QS system to autonomously and dynamically control the expression of target genes has been proven to be effective in improving production[48]. In complex metabolic networks, high-level biosynthesis of target products usually requires simultaneous regulation of multiple metabolic fluxes, which cannot be well accomplished by previously developed QS systems. In this work, we redesigned the *luxR-luxI* intergenic sequence of the lux-type QS system, offering a library of 40 QS circuit variants that can autonomously function in one cell in a pathway-independent manner. The application of these circuits enables us to autonomously and temporally regulate three metabolic fluxes involved in the SA biosynthetic pathway, yielding a titer of 2.08 g/L SA produced by engineered *E. coli* in shake flask experiments. We further used these QS systems to dynamically optimize the synthesis of 4-HC and increased its titer by tenfold, demonstrating the broad applicability of the QS systems. Our work indicates the importance of having multiple tunable dynamic circuits to balance the biosynthetic pathway. Given the rich diversity of the constructed QS library, these circuits could be used for the regulation of more complex metabolic networks.

In many expression systems, it is difficult to obtain the promoters enabling high expression levels and having low leakage[49–51]. In this work, we found that the CRP-binding site plays a significant role in controlling QS behaviors. Integration of more CRP-binding sites in the *luxR-luxI* intergenic sequence inhibits the activity of $P_{luxI}$ while strongly activating the $P_{luxR}$ transcription. The inhibition reduces leaky expression of the $P_{luxI}$ and the activation significantly expands the dynamic range of the QS system. These findings allow us to construct QS variants that possess both high dynamic ranges and low leakiness. We also found that the mutation of lux box to −10 site sequence can change the transcription level of $P_{luxI}$. Although replacing the

natural sequence with using the homologous one from various *Vibrio* species only resulted in slight differences in P$_{luxI}$ activities, this strategy made a significant contribution to expanding the diversity of the QS variants. This work elucidated how the genetic components in the promoter sequence regulate the transcription of lux-type QS system, which in principle provides molecular insights for engineering other types of QS or CRP-dependent promoters. In addition to application in biosynthesis, the QS system has also been used to probe pathogenic bacteria infections by sensing biomarkers released from infected tissues[52,53]. The regulatory mechanism and engineering strategies that we revealed and developed in this study may pave the way for developing rapid and precise QS diagnostics.

## Methods

**Experimental materials.** All plasmids and strains used in this study are summarized in Supplementary Data 1, 2, respectively. The *E. coli* strain Top10 was used as a host for plasmid construction. The *E. coli* strain BW25113 (F′) was used for fluorescence characterization and production of SA and 4-HC. The BW25113 (F′) *ppc*-knockout strain BW25113 (F′) Δ*ppc*, the *crp*-mutated strain BW25113 (F′), L159, and the *rpoA*-mutated strain BW25113 (F′) A287 were constructed by RED recombination following the standard protocols. The *E. coli* BW25113 (F′) knockout strain BW25113 (F′) Δ*ppc* was constructed by RED recombination following the standard protocols. Plasmids pZE12-luc (high-copy number), pCS27 (medium-copy number), and pSA74 (low-copy number) were used for pathway assembly. Luria-Bertani (LB) medium contains 10 g/L tryptone, 5 g/L yeast extract, and 10 g/L NaCl was used for inoculation and plasmid propagation. The modified medium M9C contains 20 g/L glycerol, 2.5 g/L glucose, 6 g/L Na$_2$HPO$_4$, 0.5 g/L NaCl, 3 g/L KH$_2$PO$_4$, 1 g/L NH$_4$Cl, 246.5 mg/L MgSO$_4$·7H$_2$O, 14.7 mg/L CaCl$_2$·2H$_2$O, 5 g/L yeast extract, 2 g/L MOPS, 0.25 mg/L CuSO$_4$·5H$_2$O, 2 mg/L vitamin B1, 1.25 mg/L H$_3$BO$_3$, 0.7 mg/L CoCl$_2$·6H$_2$O, 1.6 mg/L MnCl$_2$·4H$_2$O, 0.3 mg/L ZnSO$_4$·7H$_2$O, and 0.15 mg/L NaMoO$_4$·2H$_2$O was used for fluorescence characterization of QS circuits and microbial production of SA and 4-HC. When necessary, kanamycin, ampicillin, and chloromycetin were added to the medium at final concentrations of 50, 100, and 34 mg/L, respectively.

**DNA manipulation.** The genes *eGFP* (GenBank accession number U55762), *mCherry* (GenBank accession number MH883617), and *mOrange2* (GenBank accession number KF450807) were synthesized by Sangon Biotech company. All primers used in this study are summarized in Supplementary Data 3. To investigate the regulatory effects of the CRP-binding site on P$_{luxR}$ activity, the *luxR-luxI* intergenic sequences containing 0, 1, 2, 3, and 4 copies of the CRP-binding site were synthesized and then inserted into the pZE12-luc together with *mCherry* reporter *gene* using XbaI, BamHI and XhoI to generate plasmids pZE-P$_{0*CRP}$-*luxR-mcherry*, P$_{1*CRP}$-*luxR-mcherry*, P$_{2*CRP}$-*luxR-mcherry*, P$_{3*CRP}$-*luxR-mcherry*, and P$_{4*CRP}$-*luxR-mcherry*, respectively. To investigate the regulatory effects of CRP-binding site on P$_{luxI}$ activity, the codon-optimized *luxR* and *luxI* were synthesized by Sangon Biotech company and inserted into the pCS27 plasmid by KpnI, SalI, and HindIII to generate pCS-P$_L$lacO1-*luxR-luxI*. The *luxR-luxI* intergenic sequences containing 0, 1, 2, 3, and 4 copies of the CRP-binding site were synthesized and used to replace the P$_L$lacO1 promoter of pZE-P$_L$lacO1-*eGFP* using XhoI and KpnI to generate plasmids pZE-P$_{QS08}$-*eGFP*, pZE-P$_{QS18}$-*eGFP*, pZE-P$_{QS28}$-*eGFP*, pZE-P$_{QS38}$-*eGFP*, and pZE-P$_{QS48}$-*eGFP*, respectively. To investigate how the increased CRP-binding site numbers would control the transcription of promoter P$_{luxI}$ when both P$_{luxR}$ and P$_{luxI}$ exist in QS system, *luxR* was amplified from pCS-P$_L$lacO1-*luxR-luxI* and then inserted into the pZE-P$_{QS08}$-*eGFP*, pZE-P$_{QS18}$-*eGFP*, pZE-P$_{QS28}$-*eGFP*, pZE-P$_{QS38}$-*eGFP*, and pZE-P$_{QS48}$-*eGFP* using XbaI and AatII, resulting in pZE-*luxR*-P$_{QS08}$-*eGFP*, pZE-*luxR*-P$_{QS18}$-*eGFP*, pZE-*luxR*-P$_{QS28}$-*eGFP*, pZE-*luxR*-P$_{QS38}$-*eGFP*, and pZE-*luxR*-P$_{QS48}$-*eGFP* respectively.

The *luxR-luxI* intergenic sequence was amplified from *V. fischeri* ES114 and used to replace the P$_L$lacO1 promoter of pZE-P$_L$lacO1-*eGFP* using XhoI and KpnI to generate pZE-P$_{QS18}$-*eGFP* plasmid. Then, the lux box to −10 site sequence of pZE-P$_{QS18}$-*eGFP* was replaced by the homologous sequences from *V. fischeri* PP3, H905, EM17, ES401, MJ1, ET101, and ET401 to generate plasmids pZE-P$_{QS17}$-*eGFP*, pZE-P$_{QS16}$-*eGFP*, pZE-P$_{QS15}$-*eGFP*, pZE-P$_{QS14}$-*eGFP*, pZE-P$_{QS13}$-*eGFP*, pZE-P$_{QS12}$-*eGFP*, and pZE-P$_{QS11}$-*eGFP*, respectively.

To construct a library of QS variants, a template plasmid pZE-*luxR*-P$_{QS18}$-*luxI-eGFP* was constructed by deleting P$_L$lacO1 promoter on the plasmid pZE-luc12 and inserting *luxR*, *luxR-luxI* intergenic sequences, *luxI* and *eGFP* into it by AatII, XhoI, KpnI, EcoRI, and XbaI. Then we used eight lux box to −10 site sequences from *V. fischeri* ES114, PP3, ES401, ET401, EM17, ET101, H905, MJ1, and five P$_{luxR}$ sequences containing different CRP-binding site numbers ($n = 0, 1, 2, 3, 4$) to give a total library size of 40 combinations. To evaluate the dynamic ranges of the QS variants, the gene *luxI* was deleted in these 40 plasmids by EcoRI digestion and self-ligation. To evaluate the leaky expressions of the QS variants, the genes *luxI* and *luxR* were deleted in these 40 plasmids by EcoRI and BsaI digestions and self-ligation.

To verify the function of autonomous regulation of three sets of genes, pZE-*luxR*-P$_{QS37}$-*luxI-eGFP* plasmid from the constructed library was used as a backbone plasmid. Then, the expression cassettes P$_{QS18}$-*mCherry* and P$_{QS13}$-*mOrange2* were synthesized and inserted into the plasmid pZE-*luxR*-P$_{QS37}$-*luxI-eGFP* by AflIII, BamHI, and XbaI, yielding plasmid pZE-*luxR*-P$_{QS37}$-*luxI- eGFP* -P$_{QS18}$-*mCherry*-P$_{QS13}$-*mOrange2*. To integrate the CRISPRi system into the QS circuit, promoter P$_{QS18}$ was employed to replace two P$_{LtetO-1}$ promoters of plv-P$_{LtetO-1}$-*dCas9*-P$_{LtetO-1}$-*sgRNA* using AatII/BglII and EagI/AvrII to generate plasmid plv-P$_{QS18}$-*dCas9*-P$_{QS18}$-*sgRNA*. Then, the gene *mCherry* under the control of the constitutive promoter P$_{lpp}$ was inserted into the genome of *E. coli* strain BW25113 (F′).

For biosynthesis of SA, the expression cassette *entC-pchB* was amplified from plasmid pCS-EP[32]. The gene *eGFP* was deleted in plasmids pZE-*luxR*-P$_{QS13}$-*luxI-eGFP*, pZE-*luxR*-P$_{QS18}$-*luxI-eGFP*, pZE-*luxR*-P$_{QS33}$-*luxI-eGFP*, pZE-*luxR*-P$_{QS37}$-*luxI-eGFP* and replaced by the expression cassette *entC-pchB* using EcoRI and XbaI, to generate plasmids pZE-*luxR*-P$_{QS13}$-*luxI-entC-pchB*, pZE-*luxR*-P$_{QS18}$-*luxI-entC-pchB*, pZE-*luxR*-P$_{QS33}$-*luxI-entC-pchB*, and pZE-*luxR*-P$_{QS37}$-*luxI-entC-pchB* for SA pathway expression. The control plasmid pZE-P$_L$lacO1-*entC-pchB* was created by inserting the expression cassette *entC-pchB* into pZE12-luc by BglII and XbaI. The *ppc* downregulating plasmids plv-P$_{QS18}$-*dCas9*-P$_{QS18}$-*sg$_{ppc1}$*, plv-P$_{QS18}$-*dCas9*-P$_{QS18}$-*sg$_{ppc2}$*, plv-P$_{QS18}$-*dCas9*-P$_{QS18}$-*sg$_{ppc3}$*, and plv-P$_{QS18}$-*dCas9*-P$_{QS18}$-*sg$_{ppc4}$* targeting different positions were constructed by mutating the 20 bp target-specific sequence in the plasmid plv-P$_{QS18}$-*dCas9*-P$_{QS18}$-*sgRNA*. The expression cassette APTA was amplified from plasmid pCS-APTA[32]. It was ligated with P$_{QS13}$, P$_{QS18}$, P$_{QS33}$, or P$_{QS37}$ promoters and then inserted into the plasmid pSA74 by BsrGI, and AvrII, yielding plasmids pSA-P$_{QS13}$-APTA, pSA-P$_{QS18}$-APTA, pSA-P$_{QS33}$-APTA, and pSA-P$_{QS37}$-APTA, respectively, for enhancing the shikimate pathway expression. For dynamic modular optimization of 4-HC production, the plasmids pZE-*luxR*-P$_{QS13}$-*luxI-entC-pchB*, pZE-*luxR*-P$_{QS18}$-*luxI-entC-pchB*, pZE-*luxR*-P$_{QS33}$-*luxI-entC-pchB*, and pZE-*luxR*-P$_{QS37}$-*luxI-entC-pchB* were used for SA production. The expression cassette *pqsD-sdgA* was amplified from plasmid pCS-PS[38]. The gene *eGFP* was deleted in plasmids pZE-P$_{QS13}$-*eGFP*, pZE-P$_{QS18}$-*eGFP*, pZE-P$_{QS33}$-*eGFP*, pZE-P$_{QS37}$-*eGFP*, and replaced by the expression cassette *pqsD-sdgA* using EcoRI and AvrII, to yield plasmids pZE-P$_{QS13}$-*pqsD-sdgA*, pZE-P$_{QS18}$-*pqsD-sdgA*, pZE-P$_{QS33}$-*pqsD-sdgA*, and pZE-P$_{QS37}$-*pqsD-sdgA* for 4-HC synthesis. The expression cassettes P$_{QS13}$-*pqsD-sdgA*, P$_{QS18}$-*pqsD-sdgA*, P$_{QS33}$-*pqsD-sdgA*, and P$_{QS37}$-*pqsD-sdgA* were amplified and individually inserted into four SA production plasmids using AvrII and PciI to generate a total of 16 combinations. The static control plasmid pZE-P$_L$lacO1-EP- P$_L$lacO1-PS was constructed by inserting the expression cassette P$_L$lacO1-PS into the plasmid pZE- P$_L$lacO1-EP by using SpeI and SacI.

To construct the plasmid used for purification of CRP, *crp* gene without stop codon was amplified from the *E. coli* genome and digested by NdeI and XhoI to insert into the pET-22b, resulting in pET-CRP. The pET-LuxR plasmid used for LuxR purification was constructed by inserting the full-length *luxR* gene digested by NdeI and XhoI into the pET-22b.

**Fluorescence assays.** The *E. coli* transformants were cultivated in a 4 mL LB medium with appropriate antibiotics and grown overnight at 37 °C. Then 1 μL cultures were transferred into a black 96-well microplate with a clear bottom (Corning, NY, USA) containing 200 μL of M9C medium with appropriate antibiotics. When needed, isopropyl β-D-1-thiogalactopyranoside (IPTG) was added into the medium at 0 h with a final concentration of 0.5 mM. The plates were incubated in the Synergy HT (BioTek Instruments, VT, USA) microplate reader at 30 °C for 24 h. The OD$_{600}$ values and fluorescent intensities were quantified every 6 min. The Leica SP8 laser scanning confocal microscopes (Leica, Wetzlar, Germany) equipped with Leica DMC2900 imaging system was used to record the fluorescence of eGFP, Cherry, and mOrange2. About 200 μL of samples were taken at 7, 9, 12, and 22 h, respectively. Each sample was washed with 600 μL of phosphate-buffered saline (PBS, PH = 7.2, 2 mM KH$_2$PO$_4$, 8 mM Na$_2$HPO$_4$, 136 mM NaCl, and 2.6 mM KCl) to remove the broth. Then, it was dried in a φ 15 mm Nest confocal dish (Nest, Wuxi, China) for 5 min using the fume hood. Microscopy images were firstly taken under bright-field (objective, 40x; scale bar, 25 μm; autofocus mode; intensity, 163.7 V) to locate the cells. The GFP fluorescence intensity was detected by using an excitation filter of 488 nm and an emission filter of 525/40 nm. The mCherry fluorescence intensity was detected by using an excitation filter of 638 nm and an emission filter of 680/40 nm. The mOrange2 fluorescence intensity was detected by using an excitation filter of 552 nm and an emission filter of 566/15 nm.

**Quantification of mRNA levels.** Quantitative real-time PCR of mRNA isolated from culture samples was performed to measure the relative transcript levels of *ppc* across different strains. About 1 mL samples were collected and centrifuged at 7104×*g* for 5 min for pellet. Total mRNA was isolated by using the QIAamp RNA Blood Mini Kit (Qiagen, Duesseldorf, Germany). RNA samples were stored at −80 °C until processing. The purified mRNA was treated with RNase-free DNase I to remove any genomic DNA. The synthesis of cDNA was conducted by reverse transcription reaction with SuperScript III Reverse Transcriptase (Invitrogen, CA, USA). Then quantitative Real-Time PCR analysis was performed using the resultant cDNA by QuantStudio™ Real-Time PCR System (Thermo Fisher Scientific, MA, USA). The gene coding 16 S rRNA in *E. coli* strain BW25113 (F′) was used as the internal standard.

**Protein purification**. His-tagged CRP was expressed and purified in *E. coli*[54]. LuxR was expressed without tags and purified in *E. coli*[55]. The used RNAP was NEB® product, holoenzyme of *E. coli* RNA polymerase (Catalog #: M0551S).

**Electrophoretic mobility shift assay**. The Electrophoretic mobility shift assay (EMSA) experiments were conducted following a previously described protocol[55]. Briefly, the DNA probes were 350-bp PCR products obtained by amplifying the *luxR-luxI* intergenic sequence with 6'-FAM-tagged primers (Supplementary Table 3). Protein-DNA binding reaction mixtures finally contained the indicated concentrations of probes and protein in 20 µl buffer (0.5 mM EDTA, 15 mM Tris-HCl [pH 7.4 at 22 °C], 50 mM KCl, 50 µg of bovine serum albumin/ml, 1 mM, dithiothreitol, 10 mM AHL, and 5% glycerol). The reaction mixtures were incubated at 24 °C for 30 min and then were added with loading buffer and loaded to a native 5% gel at 4 °C. After electrophoresis at 15 V/cm at 4 °C, the bands of protein-DNA complex and unbound DNA were detected and quantitated using a Tanon® 1000 M phosphor-imager (Shanghai, China).

**Atomic force microscope observation**. The AFM observation was conducted following a previously described protocol[56]. Briefly, the mica disks were pretreated with Nickel adsorption buffer (20 mM HEPES [pH 7.4 at 22 °C] and 10 mM NiCl₂) and then were bound with DNA in Nickel imaging buffer (20 mM HEPES [pH 7.4 at 22 °C] and 2 mM NiCl₂) for 30 min. Following observation by a NanoWizard® 4 XP NanoScience atomic force microscope (Bruker, CA, USA), the obtained images were processed with JPKSPM Data Processing Python 2.7.10.

**Cultivation conditions**. For SA and 4-HC production, all transformants were inoculated into 4 mL of LB medium with appropriate antibiotics and grown overnight at 37 °C with shaking at 200 rpm. Then, 0.5 mL of overnight cultures were transferred into 250-mL baffled shake flasks containing 50 mL of M9C medium with appropriate antibiotics. The cultures were inoculated at 33 °C with shaking at 200 rpm for 72 h. When needed, IPTG was added into the medium at 0 h with a final concentration of 0.5 mM. OD₆₀₀ values were measured by UV spectrophotometer (Thermo Fisher Scientific, MA, USA). Samples were taken at 72 h and centrifuged at 10,625×g for 10 min. The supernatants were filtered using a 0.22 µm polyethersulfone (PES) membrane and analyzed by high-performance liquid chromatography (HPLC).

**HPLC analysis**. The standards of SA and 4-HC were purchased from Macklin (Shanghai, China). Samples and the standards were quantified by LC-20AT HPLC (Shimadzu, Kyoto, Japan) equipped with a UV–vis detector and a reverse-phase Diamonsil C18 column (Diamonsil 5 µm, 250 mm × 4.6 mm) for SA or an ACE C18-AR column (Advanced Chromatography Technologies Ltd, 3 µm, 150 mm × 4.6 mm) for 4-HC. Solvent A was methanol (100%) and solvent B was water with 0.1% formic acid. The column temperature was maintained at 35 °C and the injection volume was 20 µL. The mobile phase was set at a flow rate of 0.8 mL/min with gradient concentration: 95 to 30% solvent B for 20 min, 30 to 95% solvent B for 5 min, and 95% solvent B for an additional 5 min. Quantification of SA and 4-HC was based on the peak areas at UV absorbance at 303 nm.

**Reporting summary**. Further information on research design is available in the Nature Research Reporting Summary linked to this article.

## Data availability

The gene sequences of *eGFP*, *mCherry*, and *mOrange2* used in this study are available in the GenBank database under accession number U55762 [https://www.ncbi.nlm.nih.gov/nuccore/1377911/], MH883617 [https://www.ncbi.nlm.nih.gov/nuccore/MH883617], and KF450807 [https://www.ncbi.nlm.nih.gov/nuccore/KF450807], respectively. Source data are provided with this paper.

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

## Acknowledgements

This work was supported by the National Key Research and Development Program of China (2018YFA0901800), the National Natural Science Foundation of China (21908003 and 22078011).

## Author contributions

C.G. and J.W. conceived the study and wrote the manuscript. C.G., Z.Y., and H.S. performed the experiments. X.Sh. and X.Su. participated in the research. J.W., Y.Y., Y.Z., and Q.Y. revised the manuscript. J.W. and Q.Y. directed the research. All authors contributed to the article and approved the submitted version.

## Competing interests

The authors declare no competing interests.
