## [Peer Review File · Nature Communications]

Redesigning regulatory components of quorum-sensing system for diverse metabolic controlReviewers' Comments:

Reviewer #1:

Remarks to the Author:

The manuscript describes the use of a series of *Vibrio* autoinducer-1 promoter sequences that are genetically modified and daisy-chained to bring about dynamic control of metabolic pathways, ultimately leading to far superior product titers.

The overall impact of the system is impressive. The authors have started out by characterizing promoters for the expression of marker genes and then moved to metabolic pathways. In the latter case, they developed tiered control systems including using CRISPRi, that autonomously coordinate flux in order to maximize yield. In all, the outcomes are nice. The writing is generally clear with some exceptions. I would suggest re-review of English. The experiments seem to have been executed well. The conclusions are, however, not fully supported by sufficient data. This reviewer has included some comments below that should be addressed prior to publication. Some of these might warrant reformulating the manuscript to include (or not) some of the key experiments.

The comments are not provided in order of importance, rather they were constructed as the reviewer went through the paper:

L49 – oversimplification of QS. Only in the case of AHLs is there direct signal binding to a regulatory protein that triggers expression.

L91 – Gene[s] expression. Also, why note that gene expression is in response to cell density...it has nothing to do with cell density. Let's move beyond the notion that cell density is a driver. It is clearly the concentration of an autoinducer relative to the cell experiencing the autoinducer.

L125 – CRP-cAMP facilitates a turn in the DNA, usually there are two sites separated by spacer sequences enabling a complete loop structure so that distant promoters can be controlled via the same dimer or tetramer of the native regulator. The manuscript seems to be based on a simple assumption that linearization of CRP will have an effect. Clearly the mechanics of CRP are missing here.

L141-143 – Spacing matters. Transcription matters. In Fig 1c, it is likely that the number of LuxI is different. Why do you think steric hindrance is a factor in c and not b?

L269 – What specifically is the "triggering OD"? How is it found? Be explicit with the data. How valid are these statistically? Give specific ranges. On what basis do you think the triggering ODs are unchanged when three are placed in the same cell? Used 0.79, 0.89 and 0.92, what were the values revealed in the actual experiment? I'd like to see the OD data. This is hidden and not clear. I'm concerned about this because OD isn't a valid "trigger", rather it is the concentration of the autoinducer, or the concentration of the autoinducer relative to the OD. Please clarify.

L280 – It would be helpful to include the maximum fluorescence values at least in the legend so that a comparison can actually be made. The current normalized plot suggests that each protein is made to the same level and this is surely not the case. Did the authors ever swap reporters in this plasmid so that one could more effectively evaluate the transcriptional control to make sure that the levels reached are not a function of the reporter? Finally, there are no concentrations of autoinducer here or anywhere in the paper. This would be helpful to attribute the conclusions based on actual inputs.

L312 – the presentation suggests that the SA titer is a function of the transcriptional regulation of the *entC* and *pchB* genes only, but it would be interesting to ensure that all other factors aren't causative (growth rate of cells, glucose levels, concentration of autoinducer, time of sample relative to the overall culture time when growth limiting nutrients are exhausted).

L318-L341 – these results were very compelling. Nice work. My one concern is that the final titer should be relative to the total substrate consumed. With this as a measure, the growth of the cells and their metabolic regulation are indeed the reason for the increased yield. If, on the other hand, the reported titers are taken at the same time but different quantities of cells, nutrients, etc. that are remaining, it is difficult to make definitive statements. That is, there might just be a switch in product dynamics, not in the final titer. Please clarify.

L424 – I am unable to find exactly what is QS13, QS18, and QS37. Yet these form the entire basis for the work. I take it these are from the different promoters in different *Vibrio*? Or are these hybrid promoters with the various numbers of CRP sites? This is confusing and clearly very important. Clarify. Discussion – While the results are compelling and the presentation is nice. The questions I have about cause and effect need to be answered.

This reviewer is also concerned that the authors have underappreciated the regulation due to CRP and the function of CRP in *E. coli*. This has to do with the ability of CRP to put 90deg bend in the DNA, which can bring repressors that form dimers into proximity for effective repression. It is not simply stacked CRP that alters transcription. There are consequences beyond simple linearization of CRP. Also, cAMP is an effector of CRP. The media and glucose relative to cells and other metabolic activity drive the level of CRP. The authors have not discussed the relative amounts of glucose in these systems, including their dynamics, and this has a huge impact on the ability of CRP to affect gene expression. The results with different numbers of CRP might be due to effects on CRP, not on the number of CRP. This was underappreciated. This is overlaid on top of the pQS13, pQS18, and pQS37 question noted above. If these these promoters do not have any CRP, then it is difficult to see how the two components (CRP and these promoters) actually work synergistically. These elements were not clear. The paper could fall apart if the CRP are not in pQS13, pQS18, and pQS37. Moreover, it would be important to give mechanistic understanding of how many CRP are best and why they are spaced the way they are. For example, do they even bind CRP? A gel shift assay would help with this. As it stands now, there are added CRP sites, but there is no evidence to indicate that they are functional.

Reviewer #2:

Remarks to the Author:

This manuscript by Ge et. al. is a synthetic biology project that seeks to manipulate the *Vibrio* LuxIR quorum sensing (QS) circuit to develop promoters that have different timing of induction and different levels of induction. The authors develop 40 unique promoters that combine different numbers of CRP boxes (0-4) and 8 -10 elements from different *Vibrio fischeri* strains. The result is a set of promoters that has divergent expression profiles. The authors then use these promoters to significantly increase synthesis of two bioproducts in *E. coli* by both upregulating and downregulating (via induction of CRISPR elements) gene expression. Although I argue that this research does not really uncover any new regulation or biological insight, I think this promoter array is a useful tool that can be used to empirically develop genetic circuits to increase production from metabolic pathways. Overall, I think the results are well supported by the data and the findings will be of broad interest. My specific comments are:

1. Line 119-It would be informative to describe the sequence of the CRP-binding site that the authors are using aligning the site in the LuxIR promoter to the consensus
2. Line 123: "has not yet been proposed"
3. Line 126-The authors hypothesize that increasing the numbers of the CRP binding sites should increase transcription, but this model does not make sense to me. The way this system works is that there are two types of CRP regulated promoters, Class 1 and Class 2. In Class 1 regulation, CRP binds upstream and interacts with the N-terminal domain of the alpha subunit of RNAP. In Class 2 regulation, CRP binds to sites overlapping the -35, -10 region and interacts with sigma 70 and/or the C-terminal domain of alpha. In either case, RNAP can only make one contact with CRP, so I do not understand how the addition of more CRP binding sites increases transcription. The author's data indicates that this is the case but seeing as they have no explanation for how or why, I think they need to remove any statements that their studies lead to a better understanding of the natural regulation of these promoters. Furthermore, the authors should describe if the CRP binding site appears to be a class 1, which is the case I think, or a class 2.
4. Line 130:At what density was this experiment done? This is critical information since the authors are describing a QS circuit (although I gather there is no LuxI in the system). If expression was

examined at one density, then such changes in fold expression could be obtained due to alteration in timing and not an increase in total transcription. For example, if the 4 CRP binding site construct was induced earliest, then at a given time point it could have the highest expression although all constructs may eventually reach the same maximum expression. The authors really need to show curves of mCherry expression over a normal bacterial growth curve to understand what these crp binding site additions are doing to luxR promoter expression.

5. Line 133: Related to my point #3, the authors claim here that they have shown that PluxR is a CRP dependent promoter. This is an overinterpretation of their results. In order to make this claim, the authors would need to explore expression in a crp mutant and show direct binding of CRP to the CRP binding site in the luxR promoter using an EMSA or related technique. I do not think such claims are important to the major findings of this paper though so I think the authors can remove them without significantly decreasing its impact or the importance of its findings.

6. Line 167: Just to be clear, the authors should also state in this sentence "and luxI expressed from a constitutive promoter".

7. Fig. 1e- An arrow typically denotes the "+1" transcription start site so the -10 lux box should be located 5' or upstream of the arrow, not the way the authors have drawn it. This comment applies to all such diagrams in the manuscript.

8. Fig. 2b- Since the timing of induction is key to the difference in these constructs, a graph of Fig. 2b zoomed in to hours 5-20 to better show the differences would be informative along with the graph that is presented.

9. Line 235: I believe "leakiness" is defined as the percent expression of the promoter when luxI and luxR are not present, but the authors should explicitly define how they are calculating this value in the text.

10. Line 306: Do the QS promoter numbers used refer to the numbers in Fig. 2b? This is unclear.

11. I don't believe Figs. 3d and 3e are discussed in the text.

12. Line 390: replace "Taking" with "Taken"

13. Figs. 4d and 4e need a better description in the legend indicating which promoters are driving which modules that are fixed and which module is being tested in the given experiment. It is described in the text but viewing the legend and figure this is not clear. A diagram as used in previous figures would be helpful too.

Reviewer #3:

Remarks to the Author:

Chang Ge et. al Investigated the regulation mechanism of luxR-luxI intergenic sequence, and created a series of QS variants that have different behaviors by mutation of the lux box to -10 site sequence and variation of the CRP-binding site numbers. so those circuits can be used to simultaneously regulate several genes. The constructed QS variants have high dynamic range, low leakiness, so those circuits can be used to autonomously and simultaneously regulate several genes, with a manner of variable-intensity. This study expands QS system that can be used to autonomously regulate metabolic flux, however, I think this article is unsuitable for publication, unless the major issues can be addressed:

Major points:

1. As the the constructed QS variants use the same inducer compound AHL, the range of triggering OD among QS systems established in this study was narrow, so the role of those circuits for sequentially regulating genes is unobvious. In addition, Chang Ge et. al dynamically downregulate ppc to balance the relationship between production and growth via the QS promoter coupled with the expression of CRISPRi system. Since the triggering time is at the early stage of growth, the timing for repressing gene expression was early and constant.

2. On the basis of Fig.1a, LuxL is followed by the fluorescent reporter gene. Is there a direct test on the expression level of the reporter gene under the whole regulation to determine the regulated result of luxL in the whole circuit? In addition, what is the regulation result of luxL in the case of inactivation

of CRP binding sites? What is the change in comparison between the two?

3. Line 139. Why does one copy CRP binding site promote PluxI transcription? This makes people feel abnormal, can you prove this result?

4. Line 230, why is the dynamic range the ratio of the highest fluorescence value to the fluorescence value without LuxI? LuxI is an essential part of the regulation of the whole element. In the absence of LuxI, only the inhibitory effect of CRP on PluxI is retained, and eGFP must be the lowest. Does the ratio at this point make sense? The dynamic range should not be the ratio of the highest fluorescence value to the lowest fluorescence value under different cell density? It's the ratio that makes mathematical sense. In addition, what is the difference between the fluorescence value in the absence of LuxI and LuxR and that in the absence of LuxI?

5. Page 7, line 218-219, it is unreasonable to use the peak fluorescence value of the QS variant that has the lowest fluorescence signal in the library as a reference value for determining the triggering times of these QS variants. I think the triggering times should be determined using respective switching points.

6. QS system from different sources has influence on dynamic range and leakage expression of components. Have you measured the binding strength of QS system from different sources with LuxR-AHL?

7. In Supplementary Fig.3b, why do SGPPC3 and SGPPC4 strains grow better than the control?

8. Is DHAP to shikimate catalyzed by one enzyme or by multiple enzymes? Why is there no dynamic regulation of this or these genes in Fig.4a?

9. P372, Fig.4d, when three plasmids were introduced into the strain, would it have any effect on the growth of the strain?

10. In the following two product applications, why QS13, QS18, QS37 and QS33 are selected to regulate genes? They are not the strongest, so what are they chosen for?

Minor points :

1. Page 7, line 230-231, please clarify this sentence.

2. In line 207, whether the lux box numbers of different sources should be added to facilitate the understanding of Figure 2. Although Figure 2d shows this point, but still have a chart 2b, 2c and the results described in the article caused problems in understanding.

3. Line 223 "We found that the triggering OD was decreased by increasing the number of CRP-binding sites, which could be resulted from the increased AHL accumulation rates." Is there a more direct experimental result to prove this?

Reviewer #4:

Remarks to the Author:

In this manuscript, Ge and co-workers describe the characterization and modification of quorum sensing regulatory mechanisms and their implementation in synthetic metabolic pathways for bioproduction of salicylic acid and 4-hydroxycoumarin. These efforts in combination with timed-downregulation of an essential gene resulted in the improvement of salicylic acid and 4-hydroxycoumarin production titer. Even though the authors did a nice work in characterizing the -10 lux region and CRP-binding site, and in putting all these together, the outcome of having multiple CRP binding sites and diversification of -10 lux region was not surprising. The application of QS-based dynamic regulation in metabolic engineering (in combination with CRISPRi) has also been demonstrated in several previous studies (Tian, et al., 2020, NAR (doi: 10.1093/nar/gkaa602); Liu et al., 2020, ACS Synthetic Biology (doi: 10.1021/acssynbio.0c00148); Dinh and Prather, 2019 (doi.org/10.1073/pnas.1911144116), PNAS ; Kim et al., 2017, Met Eng (doi: 10.1016/j.ymben.2017.11.004)). Therefore, based on this evaluation, I think the novelty of this study is insufficient for publication in Nature Communications. This manuscript may find a better home in a more field-specific journal.

Authors may consider these suggestions below before submitting their manuscript to another peer-

reviewed journal. Most comments are made to help readers understand the paper more easily.

1. L168: Please clarify this sentence. What do you mean by: "When separate characterization of PluxR and PluxI"?

2. Fig 1c, L170: Please provide statistical analysis. Are they statistically significant? What evidence do you have to support the inhibition of PluxI by multiple CRP-binding sites in Figure 1c, 1d?

3. L218-221: To help the reader, please show the growth curve data against fluorescence measurement data and show how you determined the 'triggering OD' value in the Supplementary Information.

4. Supplementary Fig. 2: Provide a schematic diagram of plasmids that you use to obtain these figures. A schematic diagram as appears in Figure 2a will help the reader to understand the results better.

5. Supp Fig 3: The caption does not match the figures.

6. Please provide mCherry, eGFP or mOrange fluorescence measurement data of strains that carry no reporter proteins as negative controls.

7. Was maturation time of the reporter proteins taken into consideration? It looks like mCherry takes longer time to mature in comparison to GFP, and mOrange takes the longest time to mature between the three.

<https://doi.org/10.1371/journal.pone.0075991>

8. Figure 3d and 3e have not been mentioned in the text? Where in the manuscript do the authors describe and explain Figure 3d and 3e?

9. In addition to titer (mg/L), please provide data in mg/g CDW and mg/L/day. The length of production experiment in this manuscript is perhaps different from the ones in published studies. Cell growth might also affect production titer. Or, better yet, display the results in mg/g of consumed glucose.

10. Fig. 4, 5: How about protein abundance? As in many metabolic engineering studies, measurement of proteins is important to understand the difference between static vs dynamic control of gene expression.

Reviewer #1 (Remarks to the Author):

The manuscript describes the use of a series of *Vibrio* autoinducer-1 promoter sequences that are genetically modified and daisy-chained to bring about dynamic control of metabolic pathways, ultimately leading to far superior product titers.

The overall impact of the system is impressive. The authors have started out by characterizing promoters for the expression of marker genes and then moved to metabolic pathways. In the latter case, they developed tiered control systems including using CRISPRi, that autonomously coordinate flux in order to maximize yield. In all, the outcomes are nice. The writing is generally clear with some exceptions. I would suggest re-review of English. The experiments seem to have been executed well. The conclusions are, however, not fully supported by sufficient data. This reviewer has included some comments below that should be addressed prior to publication. Some of these might warrant reformulating the manuscript to include (or not) some of the key experiments.

Response:

We appreciate the reviewer for reviewing our manuscript and providing the professional and constructive comments, which helped us improve the manuscript a lot. As suggested, we have carefully revised our manuscript, the detailed modifications are listed below.

The comments are not provided in order of importance, rather they were constructed as the reviewer went through the paper:

L49 – oversimplification of QS. Only in the case of AHLs is there direct signal binding to a regulatory protein that triggers expression.

Response:

Thanks for the comment. We have modified this sentence as “Once the concentration of signaling molecules reaches a certain threshold, it will bind to a regulatory protein to trigger the expression of target genes under control of the QS promoter⁶.” in line 44-46.

L91 – Gene[s] expression. Also, why note that gene expression is in response to cell density...it has nothing to do with cell density. Let's move beyond the notion that cell density is a driver. It is clearly the concentration of an autoinducer relative to the cell experiencing the autoinducer.

Response:

Thanks for the comment. We have modified the sentence as “we established a library of 40 QS circuit variants that can control gene(s) expression in response to different concentration of signaling molecules.” in line 84-86.

L125 – CRP-cAMP facilitates a turn in the DNA, usually there are two sites separated by spacer sequences enabling a complete loop structure so that distant promoters can be controlled via the same dimer or tetramer of the native regulator. The manuscript seems to be based on a simple assumption that linearization of CRP will have an effect. Clearly the mechanics of CRP are missing here.

Response:

Thanks for the reviewer's constructive comments. As suggested, we have supplemented the mechanics of CRP as "The CRP is an allosteric protein that usually binds to its signaling molecule cyclic AMP to form dimers or tetramers. This CRP-cAMP complex can bind to a recognition site in the target core promoters to assist RNA polymerase binding to the promoter²⁵." in line 115-117.

L141-143 – Spacing matters. Transcription matters. In Fig 1c, it is likely that the number of LuxI is different. Why do you think steric hindrance is a factor in c and not b?

Response:

Thanks for the reviewer's comments. Actually, since the gene *LuxI* was expressed on the same constitutive promoter, the number of LuxI is same in all five strains. In order to avoid the misunderstanding, we have modified the sentence as "The *luxR* and *luxI* genes were expressed under the control of a constitutive promoter to make sure the number of LuxR or LuxI is same in all of strains." in line 149-150.

By increasing the CRP binding sites, we observed that the transcription of P_{luxR} is activated but the transcription of P_{luxI} is inhibited. We performed extra EMSA experiments to confirm this result. As shown in the Supplementary Figure 8 and 9, the CRP protein indeed assist RNAP binding to the promoter P_{luxR} . When incorporating four CRP binding sites in the locus, the binding band of RNAP with P_{luxR} shown no significant difference compared with that of one CRP binding site in the locus. However, the binding band of LuxR and P_{luxI} was obviously shallowed. Based on these results, we proposed the hypothesis that a high local concentration of CRP-cAMP sterically hinders the binding of the LuxR-AHL complex to the promoter P_{luxI} . To make it more reasonable. we have included these results by inserting these sentence "We further performed EMSA experiments to confirm this result. As shown in the Supplementary Fig. 8 and 9, the CRP protein indeed assist RNAP binding to the promoter P_{luxR} . When incorporating four CRP binding sites in the locus, the binding band of RNAP with P_{luxR} shown no significant difference compared with that of one CRP binding site in the locus. However, the binding band of LuxR and P_{luxI} was obviously shallowed. Based on these results, we believe that a high local concentration of CRP-cAMP might sterically hinders the binding of the LuxR-AHL complex to the promoter P_{luxI} ." in line 155-161.

Supplementary Figure 8 The electrophoretic mobility shift assay for P_{luxR} -promoter probes with 1 or 4 CRP-binding sites. DNA probes at 50 nM; CRP at 250 nM; RNAP at 300 nM.

Supplementary Figure 9 The electrophoretic mobility shift assay for P_{luxI} -promoter probes with 1 or 4 CRP-binding sites. DNA probes at 50 nM; CRP at 250 nM; LuxR at 300 nM.

L269 – What specifically is the "triggering OD"? How is it found? Be explicit with the data. How valid are these statistically? Give specific ranges. On what basis do you think the triggering ODs are unchanged when three are placed in the same cell? Used 0.79, 0.89 and 0.92, what were the values revealed in the actual experiment? I'd like to see the OD data. This is hidden and not clear. I'm concerned about this because OD isn't a valid "trigger", rather it is the concentration of the autoinducer, or the concentration of the autoinducer relative to the OD. Please clarify.

Response:

Thanks for the reviewer's comments. We defined the "triggering OD" according to the previous study (Dinh and Prather, 2019 (doi.org/10.1073/pnas.1911144116), PNAS), which can be used to compare the triggering times of different QS variants. Using the peak fluorescence value of the QS variant that has the lowest fluorescence signal in the library as a reference value, the "triggering OD" was defined as the cell density corresponding to the time of fluorescence reaching the reference value. We do not think that triggering OD is unchanged in practical applications. The above triggering ODs were used as references to characterize the relative expression order of multiple genes, but not to express genes at these static triggering ODs. The OD_{600} values of the strains that having the triggering OD of 0.79, 0.89 and 0.92 in the actual experiments are 0.868, 0.964 and 1.067, respectively. The statistical analysis is based on the Student's t-test. The P-values were less than 0.05, which were 0.0073, 0.0001 and 0.0036.

L312 – the presentation suggests that the SA titer is a function of the transcriptional regulation of the *entC* and *pchB* genes only, but it would be interesting to ensure that all other factors aren't causative (growth rate of cells, glucose levels, concentration of autoinducer, time of sample relative to the overall culture time when growth limiting nutrients are exhausted).

Response:

Thanks for the reviewer's comments. To ensure that SA titer is a function of the transcriptional regulation of the *entC* and *pchB* genes only, we performed extra experiments to prove all other factors aren't causative.

Firstly, we tested the growth rates of the SA producers (Fig. R1). The results shown that there is no significant difference among all SA producers.

Figure R1 Growth rates of SA-producing strains. All data points are reported as mean \pm s.d. from three independent experiments.

Secondly, we tested the glucose consumption for all of the SA producers (Fig. R2). The results shown that there is no significant difference among all SA producers.

Figure R2 Glucose consumption of SA-producing strains. All data points are reported as mean \pm s.d. from three independent experiments.

Thirdly, as shown in the Fig. 4b, the gene (*LuxI*) responsible for producing autoinducer has been placed on the plasmid under control of a constitutive promoter. All of SA-producing strains used the same plasmid to generate autoinducer. Thus, we believe the concentration of autoinducer is no significant difference among all SA producers.

Finally, we sampled the SA titers in all SA producers at different times including 18 h, 30 h and 42 h (Fig. R3). The results shown that there is no significant difference among all SA producers.

Figure R3 The SA titers of the SA-producing strains at different times. All data points are reported as mean \pm s.d. from three independent experiments.

Taking together, we believe that SA titer is a function of the transcriptional regulation of the *entC* and *pchB* genes only, all other factors aren't causative.

L318-L341 – these results were very compelling. Nice work. My one concern is that the final titer should be relative to the total substrate consumed. With this as a measure, the growth of the cells and their metabolic regulation are indeed the reason for the increased yield. If, on the other hand, the reported titers are taken at the same time but different quantities of cells, nutrients, etc. that are remaining, it is difficult to make definitive statements. That is, there might just be a switch in product dynamics, not in the final titer. Please clarify.

Response:

Thanks for the reviewer's comments. As suggested, we provided the data for final titer relative to the total substrate consumed (Fig. R4), final titer relative to the biomass (Fig. R5) and SA titer dynamics over cultivation time (Fig. R6). The results shown that all of these data exhibited the same trend compared with the final titer we mentioned in the text. Thus, we believe that our statements in the manuscript are definitive.

Figure R4. The SA titers per consumed substrate. All data points are reported as mean \pm s.d. from three independent experiments.

Figure R5. The SA titers per OD₆₀₀. All data points are reported as mean \pm s.d. from three independent experiments.

Figure R6. The SA titers dynamics over cultivation time for all of SA-producing strains. All data points are reported as mean \pm s.d. from three independent experiments.

L424 – I am unable to find exactly what is QS13, QS18, and QS37. Yet these form the entire basis for the work. I take it these are from the different promoters in different *Vibrio*? Or are these hybrid promoters with the various numbers of CRP sites? This is confusing and clearly very important. Clarify.

Response:

Thanks for the reviewer’s comments. QS13, QS18, and QS37 are hybrid promoters. We named the circuits as “QSMN”. The “M” represents the number of CRP-binding sites. The “N” represents the source of partial promoter sequence from eight different *Vibrio* species. In order to make it easy to understand, we have put this information in the figure legends of the figure 2.

Discussion – While the results are compelling and the presentation is nice. The questions I have about cause and effect need to be answered.

This reviewer is also concerned that the authors have underappreciated the regulation due to CRP

and the function of CRP in *E. coli*. This has to do with the ability of CRP to put 90deg bend in the DNA, which can bring repressors that form dimers into proximity for effective repression. It is not simply stacked CRP that alters transcription. There are consequences beyond simple linearization of CRP.

Response:

Thanks for the reviewer's comments. In order to confirm the function of CRP, we performed extra EMSA experiments. As shown in the Supplementary Figure 5, as the number of CRP-binding sites increases, the number of CRP protein bounded to the DNA increases too. Thus, we believe that more CRP binding sites promoted the CRP protein binding to the DNA.

Supplementary Figure 5 The electrophoretic mobility shift assay for DNA probes with different numbers of CRP-binding site ($n = 0, 1, 2, 3, 4,$ and 5) binding with CRP. (a) EMSA image. DNA probes at 50 nM ; CRP at 250 nM . (b) Gray analysis based on the EMSA image. Relative binding strength was identified by the ratio of gray values of CRP+ to CRP- bottom bands (unbound DNA probes). The gray analysis was performed by ImageJ bundled with 64-bit Java 1.8.0. All data points are reported as mean \pm s.d. from three independent experiments.

We also used atomic force microscope to observe the molecules. As shown in the Supplementary Figure 6, when incorporating four CRP binding sites in the locus, the DNA and CRP protein forms obvious multimers compared with that of one CRP binding site in the locus. We think this is an interesting result and we hypothesize that the formed multimers might promote the bounded CRP to recruit more RNAP for transcription. We are planning to spend more years on studying the mechanism, since this work itself requires systematic study and takes even longer time as indicated by many related publications. Overall, the main focus of the current study is to using the QS system to regulate the metabolic pathway for bioproduction which we think is a complete research work. We hope the reviewer can understand and support our research and publication plan. We hope the reviewer can agree with us, and we still appreciate reviewer's comment.

Supplementary Figure 6 Atomic force microscope observations of 1 and 4 CRP-binding sites DNA with or without CRP.

Also, cAMP is an effector of CRP. The media and glucose relative to cells and other metabolic activity drive the level of CRP. The authors have not discussed the relative amounts of glucose in these systems, including their dynamics, and this has a huge impact on the ability of CRP to affect gene expression. The results with different numbers of CRP might be due to effects on CRP, not on the number of CRP. This was underappreciated. This is overlaid on top of the pQS13, pQS18, and pQS37 question noted above. If these promoters do not have any CRP, then it is difficult to see how the two components (CRP and these promoters) actually work synergistically. These elements were not clear. The paper could fall apart if the CRP are not in pQS13, pQS18, and pQS37.

Response:

Thanks for the reviewer's comments. We named the circuits as "QSMN". The "M" represents the number of CRP-binding sites. The "N" represents the source of partial promoter sequence from eight different *Vibrio* species. Thus, the promoter pQS13, pQS18, and pQS37 have one, one and three CRP binding sites, respectively. The EMSA experiments shown that as the number of CRP-binding sites increases, the number of CRP protein bounded to the DNA increases too (Supplementary Figure 5). We also provided the data for glucose dynamics in the SA producers. As shown in the Fig. R2, the initial glucose concentration was only 2.5g/L, and it was no longer detectable after 3 hours of fermentation. Thus, we believe that the glucose level has no significant effect on CRP.

Figure R2 Glucose consumption of SA-producing strains. All data points are reported as mean \pm s.d. from three independent experiments.

Moreover, it would be important to give mechanistic understanding of how many CRP are best and why they are spaced the way they are. For example, do they even bind CRP? A gel shift assay would help with this. As it stands now, there are added CRP sites, but there is no evidence to indicate that they are functional.

Response:

Thanks for the reviewer's comments. As suggested, we performed EMSA experiments to confirm the CRP protein binds to the DNA. As shown in the Supplementary Fig. 5a and 5b, as the number of CRP-binding sites increases, the number of CRP protein bounded to the DNA increases too, indicating that more CRP binding sites promoted the CRP protein binding to the DNA.

Supplementary Figure 5 The electrophoretic mobility shift assay for DNA probes with different numbers of CRP-binding site ($n = 0, 1, 2, 3, 4,$ and 5) binding with CRP. (a) EMSA image. DNA probes at 50 nM ; CRP at 250 nM . (b) Gray analysis based on the EMSA image. Relative binding strength was identified by the ratio of gray values of CRP+ to CRP- bottom bands (unbound DNA probes). The gray analysis was performed by ImageJ bundled with 64-bit Java 1.8.0. All data points are reported as mean \pm s.d. from three independent experiments.

We also performed extra experiments to investigate how many CRP are best and why they are spaced the way they are. As shown in the Supplementary Figure 4, when the CRP binding site varies from 0 to 7, the QS promoter harboring 4 CRP binding sites exhibited the highest transcription level and dynamic range, indicating that 4 CRP binding sites is the best.

Supplementary Figure 4 Gene expression of the QS variants with tandem copies of CRP-binding site ($n=0,1,2,3,4,5,6,7$). All data points are reported as mean \pm s.d. from three independent experiments.

We also used atomic force microscope to observe the topographies of CRP-DNA complex. As shown in the Supplementary Fig. 6, when incorporating four CRP binding sites in the locus, the

DNA and CRP protein forms obvious multimers compared with that of one CRP binding site in the locus. We believe that the formed multimers might promote the bounded CRP to recruit more RNAP for transcription.

Supplementary Figure 6 Atomic force microscope observations of 1 and 4 CRP-binding sites DNA with or without CRP.

Reviewer #2 (Remarks to the Author):

This manuscript by Ge et. al. is a synthetic biology project that seeks to manipulate the Vibrio LuxIR quorum sensing (QS) circuit to develop promoters that have different timing of induction and different levels of induction. The authors develop 40 unique promoters that combine different numbers of CRP boxes (0-4) and 8 -10 elements from different Vibrio fischeri strains. The result is a set of promoters that has divergent expression profiles. The authors then use these promoters to significantly increase synthesis of two bioproducts in E. coli by both upregulating and downregulating (via induction of CRISPR elements) gene expression. Although I argue that this research does not really uncover any new regulation or biological insight, I think this promoter array is a useful tool that can be used to empirically develop genetic circuits to increase production from metabolic pathways. Overall, I think the results are well supported by the data and the findings will be of broad interest. My specific comments are:

1. Line 119-It would be informative to describe the sequence of the CRP-binding site that the authors are using aligning the site in the LuxIR promoter to the consensus.

Response:

We appreciate the reviewer for reviewing our manuscript and providing the professional and constructive comments, which helped us improve the manuscript a lot. As suggested, we have provided the consensus sequence of the CRP binding site in the manuscript in line 114. Additionally, we provided the sequence alignment results in the Supplementary Figure 1.

2. Line 123: “has not yet been proposed”

Response:

Thanks for the reviewer’s comments. We have corrected this sentence as “**However, the exact function... has not yet been proposed.**” in line 118.

3. Line 126-The authors hypothesize that increasing the numbers of the CRP binding sites should increase transcription, but this model does not make sense to me. The way this system works is that there are two types of CRP regulated promoters, Class 1 and Class 2. In Class 1 regulation, CRP binds upstream and interacts with the N-terminal domain of the alpha subunit of RNAP. In Class 2 regulation, CRP binds to sites overlapping the -35, -10 region and interacts with sigma 70 and/or the C-terminal domain of alpha. In either case, RNAP can only make one contact with CRP, so I do not understand how the addition of more CRP binding sites increases transcription. The author’s data indicates that this is the case but seeing as they have no explanation for how or why, I think they need to remove any statements that their studies lead to a better understanding of the natural regulation of these promoters. Furthermore, the authors should describe if the CRP binding site appears to be a class 1, which is the case I think, or a class 2.

Response:

Thanks for the reviewer’s comments. In order to confirm that increasing the numbers of the CRP binding sites should increase transcription, we performed extra EMSA experiments. As shown in the Supplementary Figure 5, as the number of CRP-binding sites increases, the number of CRP protein bounded to the DNA increases too. Thus, we believe that more CRP binding sites promoted the CRP protein binding to the DNA.

Supplementary Figure 5 The electrophoretic mobility shift assay for DNA probes with different numbers of CRP-binding site ($n = 0, 1, 2, 3, 4,$ and 5) binding with CRP. (a) EMSA image. DNA probes at 50 nM ; CRP at 250 nM . (b) Gray analysis based on the EMSA image. Relative binding strength was identified by the ratio of gray values of CRP+ to CRP- bottom bands (unbound DNA probes). The gray analysis was performed by ImageJ bundled with 64-bit Java 1.8.0. All data points are reported as mean \pm s.d. from three independent experiments.

We also used atomic force microscope to observe the molecules. As shown in the Supplementary Figure 6, when incorporating four CRP binding sites in the locus, the DNA and CRP protein forms obvious multimers compared with that of one CRP binding site in the locus. We think this is an interesting result and we believe that the formed multimers might promote the bounded CRP to recruit more RNAP for transcription.

Supplementary Figure 6 Atomic force microscope observations of 1 and 4 CRP-binding sites DNA with or without CRP.

According to the previous report, the Leu159 substitution in CRP and Ala287 substitution in RNAP can repress class I CRP-dependent activation^{1,2} As shown in the Supplementary Figure 7, the obvious decreases in transcription led by carrying out these two substitutions indicates that the CRP binding site appears to be a class I.

Supplementary Figure 7 The expression of P_{luxR} after performing the Leu 159 substitution in CRP or the Ala 287 substitution in the α CTD of RNAP. These two substitutions are especially harmful to class I CRP-dependent activation, which were used to identify class I CRP-dependent activation. **(a)** Schematic diagram of the residues substituted in the protein. AR 1, activating region 1. **(b)** The expression of P_{luxR} after performing the Leu 159 substitution or the Ala 287 substitution. All data points are reported as mean \pm s.d. from three independent experiments.

We have inserted these sentences to describe them “We further performed electrophoretic mobility shift assay (EMSA) experiments to confirm the result. As shown in the Supplementary Fig. 5a and 5b, as the number of CRP-binding sites increases, the number of CRP protein bounded to the DNA increases too, indicating that more CRP binding sites promoted the CRP protein binding to the DNA. We also used atomic force microscope to observe the topographies of CRP-DNA complex. As shown in the Supplementary Fig. 6, when incorporating four CRP binding sites in the locus, the DNA and CRP protein forms obvious multimers compared with that of one CRP binding site in the locus. We believe that the formed multimers might promote the bounded CRP to recruit more RNAP for transcription. Since P_{luxR} is a type of CRP regulated promoter, it is necessary to further explore which class it belongs to. The amino acid H159 in the CRP protein was substituted to Leu and the amino acid V287 in the alpha subunit of RNAP was substituted to Ala (Supplementary Fig. 7a and 7b). The results shown that both substitutions dramatically reduced the transcription of P_{luxR} (Supplementary Fig. 7c), indicating that P_{luxR} appears to be a class I CRP-dependent promoter.” in line 133-146.

4. Line 130: At what density was this experiment done? This is critical information since the authors are describing a QS circuit (although I gather there is no LuxI in the system). If expression was examined at one density, then such changes in fold expression could be obtained due to alteration in timing and not an increase in total transcription. For example, if the 4 CRP binding site construct was induced earliest, then at a given time point it could have the highest expression although all constructs may eventually reach the same maximum expression. The authors really need to show curves of mCherry expression over a normal bacterial growth curve to understand what these crp binding site additions are doing to luxR promoter expression.

Response:

Thanks for the reviewer's comments. As suggested, we provided the curves of mCherry expression over a normal bacterial growth curve in the Fig. R7. The results shown that there is no significant difference in growth curve of all strains. The changes in fold expression were caused by the increase in total transcription not by the alteration in timing.

Figure R7 The mCherry expression of P_{luxR} with different copies of CRP-binding site over a normal bacterial growth. All data points are reported as mean from three independent experiments.

5. Line 133: Related to my point #3, the authors claim here that they have shown that PluxR is a CRP dependent promoter. This is an overinterpretation of their results. In order to make this claim, the authors would need to explore expression in a *crp* mutant and show direct binding of CRP to the CRP binding site in the *luxR* promoter using an EMSA or related technique. I do not think such claims are important to the major findings of this paper though so I think the authors can remove them without significantly decreasing its impact or the importance of its findings.

Response:

Thanks for the reviewer's comments. As suggested, we have removed this claim in line 128.

6. Line 167: Just to be clear, the authors should also state in this sentence “and *luxI* expressed from a constitutive promoter”.

Response:

Thanks for the reviewer's comments. As suggested, we have modified this sentence as "This result indicates that an increase of the CRP-binding site numbers could significantly expand the dynamic range of P_{luxI} when co-expressing the P_{luxR} controlled *luxR* in the system and *luxI* expressed from a constitutive promoter." in line 181-184.

7. Fig. 1e-An arrow typically denotes the "+1" transcription start site so the -10 lux box should be located 5' or upstream of the arrow, not the way the authors have drawn it. This comment applies to all such diagrams in the manuscript.

Response:

Thanks for the reviewer's comments. As suggested, we have corrected this mistake in the figure 1 and figure 2.

8. Line 235: I believe "leakiness" is defined as the percent expression of the promoter when *luxI* and *luxR* are not present, but the authors should explicitly define how they are calculating this value in the text.

Response:

Thanks for the reviewer's comments. As suggested, we have supplemented this information by inserting the sentence "Thus, we also evaluated the expression leakage of these QS variants, which was calculated as the ratio of comparing peak fluorescence intensity from the strains carrying partial QS circuits in the absence of *luxR* and *luxI* to the strains harboring entire QS circuits in the presence of *luxR* and *luxI*." in line 256-259.

9. Line 306: Do the QS promoter numbers used refer to the numbers in Fig. 2b? This is unclear.

Response:

Thanks for the reviewer's comments. Yes, the QS promoter numbers used refer to the numbers in Fig. 2b. To make it clear, we have supplemented this information by inserting the sentence "To test this hypothesis, QS promoters P_{QS13} , P_{QS18} , P_{QS37} and P_{QS33} from the library (Fig. 2b) were individually employed for driving the expression of *entC* and *pchB*, ..." in line 338-340.

10. I don't believe Figs. 3d and 3e are discussed in the text.

Response:

We appreciate the reviewer's comments. We have corrected this mistake and described the Figure 3d and 3e by inserting these sentences "The above-established QS circuits only have the function of activating transcription. To expand the functionality of the QS system, we tested the potential of using CRISPRi as a switch to achieve signal conversion. Thus, the AHL-induced up-regulation was transduced into a down-regulation function by integrating the CRISPRi system into QS circuits. To confirm whether simultaneously dynamic up-regulation and down-regulation can be achieved in an orthogonal manner, we constructed plasmid $pZE-luxR-P_{QS18-luxI-eGFP}$ as the reporter of the up-regulation function. To construct the down-regulation function reporter, the plasmid $plv-P_{QS18-dCas9-sgRNA}$ was constructed to repress the *mCherry* expression on the genome of *E. coli* BW25113 (F') (Fig. 3d). As seen from Fig. 3e, in the presence of the AHL and CRISPRi system, the eGFP fluorescence intensity was up-regulated by 7.7-fold, and the

mCherry fluorescence intensity was down-regulated by 82.9% by the end of 24 h. This result demonstrates that coupling the CRISPRi system with the QS variants could generate a dynamic and bi-functional regulation tool for simultaneously up- and down-regulate two sets of genes.” in line 306-318.

11. Line 390: replace “Taking” with “Taken”

Response:

We appreciate the reviewer’s comments. We have corrected this mistake by replacing the “Taking” with “Taken” in line 426.

12. Figs. 4d and 4e need a better description in the legend indicating which promoters are driving which modules that are fixed and which module is being tested in the given experiment. It is described in the text but viewing the legend and figure this is not clear. A diagram as used in previous figures would be helpful too.

Response:

We appreciate the reviewer’s comments. As suggested, we have inserted these sentences “The up-regulation module containing SA pathway genes was controlled by the QS variant P_{QS37} , the down-regulation module carrying CRISPRi system was controlled by the QS variant P_{QS18} , the APTA module was controlled by QS variant P_{QS13} , P_{QS18} , P_{QS37} or P_{QS33} , respectively.” in the legends of the fig. 4d and fig. 4e.

Reviewer #3 (Remarks to the Author):

Chang Ge et. al Investigated the regulation mechanism of luxR-luxI intergenic sequence, and created a series of QS variants that have different behaviors by mutation of the lux box to -10 site sequence and variation of the CRP-binding site numbers. so those circuits can be used to simultaneously regulate several genes. The constructed QS variants have high dynamic range, low leakiness, so those circuits can be used to autonomously and simultaneously regulate several genes, with a manner of variable-intensity. This study expands QS system that can be used to autonomously regulate metabolic flux, however, I think this article is unsuitable for publication, unless the major issues can be addressed:

Response:

We appreciate the reviewer for reviewing our manuscript and providing the professional and constructive comments, which helped us improve the manuscript a lot.

Major points:

1. As the constructed QS variants use the same inducer compound AHL, the range of triggering OD among QS systems established in this study was narrow, so the role of those circuits for sequentially regulating genes is un conspicuous. In addition, Chang Ge et. al dynamically downregulate *ppc* to balance the relationship between production and growth via the QS promoter coupled with the expression of CRISPRi system. Since the triggering time is at the early stage of growth, the timing for repressing gene expression was early and constant.

Response:

Thanks for the reviewer's comments. Actually, we used the microplate reader to continuously record the bacterial fluorescence and corresponding OD₆₀₀. As shown in the fig. R7, the OD₆₀₀ values of the bacterial growth measured by the microplate reader range from 0.1 to 1.7. While, the triggering ODs of QS variants in the library range from 0.572 to 1.437. This range covers 54.06% of the OD interval of the bacterial growth curve, from the early period of the log phase to the early period of the stationary phase. Therefore, we don't agree with the reviewer that the triggering ODs among QS systems established in this study was narrow and the timing for repressing gene expression was early. As shown in the fig. 3b and 3c, when using three QS variants with different triggering OD to control the expression of three different reporter genes *egfp*, *mCherry* and *mOrange2*, the green, red and yellow fluorescence was sequentially observed in the fluorescent images, indicating that the role of those circuits for sequentially regulating genes is conspicuous.

Figure R7 The mCherry expression of P_{luxR} with different copies of CRP-binding site over a normal bacterial growth. All data points are reported as mean from three independent experiments.

2. On the basis of Fig.1a, LuxL is followed by the fluorescent reporter gene. Is there a direct test on the expression level of the reporter gene under the whole regulation to determine the regulated result of luxL in the whole circuit? In addition, what is the regulation result of luxL in the case of inactivation of CRP binding sites? What is the change in comparison between the two?

Response:

Thanks for the reviewer's comments. As shown in the figure 2, the expression profile of fluorescent reporter gene eGFP placed downstream of P_{luxI} and luxI recorded the whole regulation result in the whole circuit (Fig. 2a). The QS variants exhibited a wide range of peak fluorescence intensities and triggering times (Fig. 2b and 2c). The regulation result of LuxI in the case of inactivation of CRP binding sites was also demonstrated in the Fig. 2b and 2c. When the CRP binding site is absent in the circuit ($n=0$), the maximum fluorescence intensity of the strains was dramatically decreased and the triggering ODs was increased. In addition, compared with in the presence of CRP binding site(s), inactivation of CRP-binding site resulting in lower dynamic ranges (Figure 2e) and higher leaky expressions (Figure 2f).

3.Line 139, Why does one copy CRP binding site promote P_{luxI} transcription? This makes people feel abnormal, can you prove this result?

Response:

Thanks for the reviewer's comments. According to the previous report, the Leu159 substitution in CRP and Ala287 substitution in RNAP can repress class I CRP-dependent activation^{1, 2}. As

shown in the Supplementary Figure 7, the obvious decreases in transcription led by carrying out these two substitutions indicates that the CRP binding site appears to be a class I.

Supplementary Figure 7 The expression of P_{luxR} after performing the Leu 159 substitution in CRP or the Ala 287 substitution in the α CTD of RNAP. These two substitutions are especially harmful to class I CRP-dependent activation, which were used to identify class I CRP-dependent activation. (a) Schematic diagram of the residues substituted in the protein. AR 1, activating region 1. (b) The expression of P_{luxR} after performing the Leu 159 substitution or the Ala 287 substitution. All data points are reported as mean \pm s.d. from three independent experiments.

Additionally, we preformed extra EMSA experiments and the results shown that CRP helps RNAP to bind promoters (Supplementary Figure 5). These results indicated that when the number of CRP-binding site increases from 0 to 1, the promotion effect on P_{luxI} transcription was caused by the role of CRP in recruiting RNAP, which is necessary for transcription.

Supplementary Figure 5 The electrophoretic mobility shift assay for DNA probes with different numbers of CRP-binding site ($n = 0, 1, 2, 3, 4,$ and 5) binding with CRP. (a) EMSA image. DNA probes at 50 nM ; CRP at 250 nM . (b) Gray analysis based on the EMSA image. Relative binding strength was identified by the ratio of gray values of CRP+ to CRP- bottom bands (unbound DNA probes). The gray analysis was performed by ImageJ bundled with 64-bit Java 1.8.0. All data points are reported as mean \pm s.d. from three independent experiments.

4. Line 230, why is the dynamic range the ratio of the highest fluorescence value to the

fluorescence value without LuxI? LuxI is an essential part of the regulation of the whole element. In the absence of LuxI, only the inhibitory effect of CRP on PluxI is retained, and eGFP must be the lowest. Does the ratio at this point make sense? The dynamic range should not be the ratio of the highest fluorescence value to the lowest fluorescence value under different cell density? It's the ratio that makes mathematical sense. In addition, what is the difference between the fluorescence value in the absence of LuxI and LuxR and that in the absence of LuxI?

Response:

Thanks for the reviewer's comments. We calculated the dynamic range based on the previous studies (Chen, Y. et al. Nat. Commun. 9, 1-8 (2018). and Wu, J. et al. Nat. Commun. 11, 1-14 (2020).). The LuxI produces the autoinducer AHL to induce the QS system to "ON" state, while remove of LuxI leads the QS system "OFF" state. Actually, deletion of *luxI* was commonly used to characterize the QS systems in the OFF state⁵⁻⁶. Thus, the ratio of highest fluorescence value with entire elements (ON state) to the fluorescence value without LuxI (OFF state) was used to present the "dynamic range". Additionally, the fluorescence values in the absence of LuxI are higher than that in the absence of LuxI and LuxR, which might be caused by the leaky expression induced by LuxR.

5. Page 7, line 218-219, it is unreasonable to use the peak fluorescence value of the QS variant that has the lowest fluorescence signal in the library as a reference value for determining the triggering times of these QS variants. I think the triggering times should be determined using respective switching points.

Response:

Thanks for the reviewer's comments. We defined the "triggering OD" according to the previous study (Dinh and Prather, 2019 (doi.org/10.1073/pnas.1911144116), PNAS), which can be used to compare the triggering times of different QS variants. The triggering OD was well-matched with the switching point. For example, the switch points of the strains that having the triggering OD of 0.79, 0.89 and 0.92 in the experiments are 0.868, 0.964 and 1.067, respectively.

6. QS system from different sources has influence on dynamic range and leakage expression of components. Have you measured the binding strength of QS system from different sources with LuxR-AHL?

Response:

Thanks for the reviewer's comments. As suggested, we have performed extra EMSA experiments to measure the binding strength of QS system from different sources with LuxR-AHL. As shown in the Supplementary Fig. 10, use of the homologous sequence to replace the natural one (ES114) decreased the relative binding strength of the QS system. The trend was well-matched with their transcriptional intensities.

Supplementary Figure 10 The electrophoretic mobility shift assay for QS systems from different sources with LuxR-AHL. (a) EMSA image. DNA probes at 20 nM; LuxR at 300 nM; RNAP at 300 nM. (b) Gray analysis based on the EMSA image. Relative binding strength was identified by the ratio of gray values of LuxR+ to LuxR- bottom bands (unbound DNA probes). The gray analysis was performed by ImageJ bundled with 64-bit Java 1.8.0. All data points are reported as mean \pm s.d. from three independent experiments.

We have inserted these sentences in the manuscript to describe it “The EMSA experiments were carried out to test the relative binding strength of QS system from different sources with LuxR-AHL (Supplementary Fig. 10a). The results shown that use of the homologous sequence to replace the natural one (ES114) decreased the relative binding strength of the QS system. The trends were well-matched with their transcriptional intensities (Supplementary Fig. 10b).” in line 210-214.

7. In Supplementary Fig.3b, why do SGPPC3 and SGPPC4 strains grow better than the control?

Response:

Thanks for the reviewer’s comments. The control strain was constructed by completely deleting the *ppc* gene. Since this gene is essential for cell growth, the control strain exhibited poor cell growth. On the contrary, the *ppc* gene was retained in the strains SGPPC3 and SGPPC4, but introduced the CRISPRi system (driven by the QS system) to down-regulate the gene *ppc*. In this condition, the expression of the gene *ppc* was repressed until the cell density reaching to a threshold level. Therefore, the strains SGPPC3 and SGPPC4 grown better than the control strain.

8. Is DHAP to shikimate catalyzed by one enzyme or by multiple enzymes? Why is there no dynamic regulation of this or these genes in Fig.4a?

Response:

Thanks for the reviewer’s comments. In *E. coli*, the DHAP to shikimate is catalyzed by three enzymes, AroB, AroD and AroE. We selected genes *aroG*, *ppsA*, *iktA* and *aroL* as the targets, because these genes are key factors for high-level production of the shikimate pathway derived products^{7, 8}. The 3-deoxy-D-arabino-heptulosonate-7-phosphate synthase in the shikimate pathway encoded by *aroG* is feedback-inhibited by phenylalanine (Kikuchi, Tsujimoto, & Kurahashi, 1997). The limited supply of PEP and erythrose-4-phosphate (E4P) can be eliminated by over-expression of PEP synthase (encoded by *ppsA*) and transketolase (encoded by *iktA*) (Jiang & Zhang, 2016), respectively. The shikimate kinase in the shikimate pathway was another

bottleneck, which can be alleviated by overexpression of *aroL* (DeFeyter & Pittard, 1986). Then, we selected *entC* and *pchB* as the targets because they are essential genes responsible for SA production. Finally, we selected *ppc* as the target gene for down-regulation, because this strategy has been proven by our group that is effective on enhancing the PEP availability for SA over-production (Yang, Y. et al. Nat. Commun. 9, 3043 (2018)).

9. P372, Fig.4d, when three plasmids were introduced into the strain, would it have any effect on the growth of the strain?

Response:

Thanks for the reviewer's comments. We have provided the cell growth of the strains in the Fig. 4d. As shown in the Figure R8, the differences of the OD₆₀₀ values among these strains were below 20%, indicating that introduction of three plasmids has no significant impact on the cell growth.

Figure R8 Growth profiles for 2-plasmids strain and 3-plasmids strains. The 2-plasmids strain is the top SA-producing strain using two-layered dynamic control strategy. The 3-plasmids strains were constructed by transforming the third-layered APTA module controlled by four different QS promoters into the above 2-plasmids strain. All data points are reported as mean \pm s.d. from three independent experiments.

10. In the following two product applications, why QS13, QS18, QS37 and QS33 are selected to regulate genes? They are not the strongest, so what are they chosen for?

Response:

Thanks for the reviewer's comments. We selected QS13, QS18, QS37 and QS33 to regulate genes for SA and 4-HC production because these circuits were well-characterized in the former section (line 259-295, Fig. 3a-e) in our manuscript.

Minor points:

1. Page 7, line 230-231, please clarify this sentence.

Response:

We appreciate the reviewer's comments. As suggested, this sentence has been modified as "The dynamic range is defined as the ratio of peak GFP fluorescence value of the strain carrying entire QS elements to that of the strain harboring partial QS circuits lacking the *luxI* gene." in line 250-251.

2. In line 207, whether the lux box numbers of different sources should be added to facilitate the understanding of Figure 2. Although Figure 2d shows this point, but still have a chart 2b, 2c and the results described in the article caused problems in understanding.

Response:

We appreciate the reviewer's comments. As suggested, we have modified this sentence as "By altering the numbers of the CRP-binding site ($n=0,1,2,3,4$) and the sequence of the lux box to -10 site region (ES114(8), PP3(7), H905(6), EM17(5), ES401(4), MJ1(3), ET101(2) and ET401(1)), a total of 40 QS variants were generated." in line 226-228.

3. Line 223 "We found that the triggering OD was decreased by increasing the number of CRP-binding sites, which could be resulted from the increased AHL accumulation rates." Is there a more direct experimental result to prove this?

Response:

We appreciate the reviewer's comments. As suggested, we have tested the AHL accumulation rates of the QS systems harboring different CRP binding sites ($n=0, 1, 2, 3, 4$). As shown in the Fig. R9, the AHL accumulation rate was increased by increasing the number of CRP-binding sites, indicating that the decreased triggering OD was resulted from the increased AHL accumulation rates.

Figure R9 AHL profiles for the QS systems with different numbers of CRP-binding site. (a) The corresponding relationship between AHL and relative fluorescence intensity. (b) The AHL accumulation rates of QS systems with tandem copies of CRP-binding site ($n=0,1,2,3,4$). All data points are reported as mean \pm s.d. from three independent experiments.

Reviewer #4 (Remarks to the Author):

In this manuscript, Ge and co-workers describe the characterization and modification of quorum sensing regulatory mechanisms and their implementation in synthetic metabolic pathways for bioproduction of salicylic acid and 4-hydroxycoumarin. These efforts in combination with timed-downregulation of an essential gene resulted in the improvement of salicylic acid and 4-hydroxycoumarin production titer. Even though the authors did a nice work in characterizing the -10 lux region and CRP-binding site, and in putting all these together, the outcome of having multiple CRP binding sites and diversification of -10 lux region was not surprising. The application of QS-based dynamic regulation in metabolic engineering (in combination with CRISPRi) has also been demonstrated in several previous studies (Tian, et al., 2020, NAR (doi: 10.1093/nar/gkaa602); Liu et al., 2020, ACS Synthetic Biology (doi: 10.1021/acssynbio.0c00148); Dinh and Prather, 2019 (doi.org/10.1073/pnas.1911144116), PNAS ; Kim et al., 2017, Met Eng (doi: 10.1016/j.ymben.2017.11.004)). Therefore, based on this evaluation, I think the novelty of this study is insufficient for publication in Nature Communications. This manuscript may find a better home in a more field-specific journal.

Response:

We thank the reviewer's comments. However, we do not agree with the reviewer. We think the reviewer overlooked the significance and novelty of our work for the following reasons:

First, the reviewer stated that the outcome of having multiple CRP binding sites and diversification of -10 lux region was not surprising is not justified. We reported for the first time that integration of more CRP-binding sites in the luxR-luxI intergenic sequence inhibits the activity of P_{luxI} while strongly activates the P_{luxR} transcription. We reported for the first time that the inhibition reduces leaky expression of the P_{luxI} and the activation significantly expands the dynamic range of the QS system. We reported for the first time that the mutation of lux box to -10 site sequence can change the transcription level of P_{luxI} which made a significant contribution to expanding the diversity of the QS variants. Our work elucidated for the first time that how the genetic components in the promoter sequence regulate the transcription of lux-type QS system, which in principle provides molecular insights for engineering other types of QS or CRP-dependent promoters.

Second, we agree with the reviewer that the application of QS-based dynamic regulation in metabolic engineering (in combination with CRISPRi) has been demonstrated in several previous studies. However, our work is totally different from these studies. Our work discovered regulatory roles of a CRP-binding site and the lux box to -10 region within luxR-luxI intergenic sequence in controlling the lux-type QS promoters, which have never been reported before, therefore, our work is innovative. Based on these findings, we created a library of QS variants that possess both high dynamic ranges and low leakiness, which have never been achieved before. Furthermore, we demonstrated that our created QS variants can simultaneously regulate the expression of three or more sets of genes in one cell, which have never been achieved before. In addition, we demonstrated that the QS circuits can be used for dynamic modular optimization of pathway genes, which have never been achieved before.

Overall, we think this work is of great significance, of great practical value, and innovative. We hope the reviewer can agree with us and we still appreciate the reviewer's comments.

Authors may consider these suggestions below before submitting their manuscript to another peer-reviewed journal. Most comments are made to help readers understand the paper more easily.

1. L168: Please clarify this sentence. What do you mean by: "When separate characterization of PluxR and PluxI"?"

Response:

We thank the reviewer's comments. We have modified this sentence as "We found that integration of more CRP-binding sites in the *luxR-luxI* intergenic sequence inhibits the transcription of P_{luxI} (in the absence of P_{luxR}) but significantly enhances the transcription of P_{luxR} (in the absence of P_{luxI})." in line 184-186 to make it clear.

2. Fig 1c, L170: Please provide statistical analysis. Are they statistically significant? What evidence do you have to support the inhibition of PluxI by multiple CRP-binding sites in Figure 1c, 1d?

Response:

We thank the reviewer's comments. We have provided the Student's t-test for the data in the Fig. 1c. The $*P < 0.05$ and $**P < 0.025$. They are statistically significant. We further performed electrophoretic mobility shift assay (EMSA) experiments to confirm the result. As shown in the Supplementary Fig. 9, when incorporating four CRP binding sites in the locus, the binding band of LuxR and P_{luxI} was obviously shallowed, indicating that increasing CRP-binding sites inhibited the transcription of the P_{luxI} .

Numbers of CRP-binding site	1	4	1	4	1	4	1	4
LuxR	-	-	-	-	+	+	+	+
CRP	-	-	+	+	-	-	+	+

Supplementary Figure 9 The electrophoretic mobility shift assay for P_{luxI} -promoter probes with 1 or 4 CRP-binding sites. DNA probes at 50 nM; CRP at 250 nM; LuxR at 300 nM.

3. L218-221: To help the reader, please show the growth curve data against fluorescence measurement data and show how you determined the ‘triggering OD’ value in the Supplementary Information.

Response:

We thank the reviewer’s comments. As suggested, we have provided the growth curve data against fluorescence measurement data and how we determined the ‘triggering OD’ in the Supplementary Fig. R10.

$$\text{Triggering OD (PQS18)} = (0.875 + 0.898 + 0.901) / 3 = 0.891$$

Supplementary R10 Schematic diagram of determining the “triggering OD”. (a)(b)(c) Three independent experiments of determining the “triggering OD” for P_{Qs18}. (d) The calculation of “triggering OD” exhibited in manuscript.

4. Supplementary Fig. 2: Provide a schematic diagram of plasmids that you use to obtain these

figures. A schematic diagram as appears in Figure 2a will help the reader to understand the results better.

Response:

We thank the reviewer's comments. As suggested, we have added a schematic diagram in the Supplementary Fig. 2a.

5. Supp Fig 3: The caption does not match the figures.

Response:

We thank the reviewer's comments. We have corrected this mistake and modified the caption of the Supplementary Fig. 3 as "Supplementary Figure 3 The effect of simultaneous and dynamic up-regulation of *entC-pchB* and down-regulation of *ppc* on SA production. (a) The effect of different sgRNA targeting different locations of *ppc* operon on the transcriptional level of *ppc*. The expression level of *ppc* in all strains was normalized by that of the control strain. Control represents the transcriptional level of *ppc* in the strain without *ppc* down-regulation. N. D. = Not Detected. (b) Growth profiles for all SA-producing strains. All data points are reported as mean \pm s.d. from three independent experiments."

6. Please provide mCherry, eGFP or mOrange fluorescence measurement data of strains that carry no reporter proteins as negative controls.

Response:

We thank the reviewer's comments. As suggested, we have provided the fluorescence measurement data of negative control that carries no reporter protein in the Supplementary Fig. R11.

Supplementary Figure R11 The Cherry, eGFP or mOrange fluorescence measurement data of the negative control strain that carry no reporter proteins. All data points are reported as mean \pm s.d. from three independent experiments.

7. Was maturation time of the reporter proteins taken into consideration? It looks like mCherry takes longer time to mature in comparison to GFP, and mOrange takes the longest time to mature between the three.

Response:

We thank the reviewer's comments. We performed extra experiments to confirm the sequential expression is not caused by the different maturation time of the reporter proteins. We placed the *eGFP* under control of the QS promoters P_{QS37} , P_{QS13} and P_{QS18} by constructing plasmids $_{QS37}$ -*eGFP*-CO, $_{QS13}$ -*eGFP*-CO and $_{QS18}$ -*eGFP*-CO, respectively. We tested the fluorescence profiles of the strains harboring these circuits. As shown in the Fig. R12, when using QS promoters P_{QS37} , P_{QS13} and P_{QS18} to drive the expression of a same reporter gene, the expression sequence is consistent with that in the fig. 3, indicating that the sequential expression is not caused by the different maturation time of the reporter proteins.

Figure R12 Autonomous and simultaneous control of three different gene targets by QS variants. (a) Architecture of the QS circuits that simultaneously regulating the expression of eGFP, mCherry and mOrange2. Specifically, all promoters, P_{QS13} , P_{QS18} and P_{QS37} , expressed with eGFP, attempting to eliminating the potential impact of different protein maturation times. P_{QS13} -*eGFP*-CO was used to present P_{QS13} -controlled eGFP expression. P_{QS18} -*eGFP*-CO was used to present P_{QS18} -controlled eGFP expression. P_{QS37} -*eGFP*-CO was used to present P_{QS37} -controlled eGFP expression. (b) Fluorescence profiles for eGFP in the strains containing the designed QS circuit. For all strains, the fluorescent intensity of each time point was normalized by their corresponding peak fluorescence value. All data points are reported as mean \pm s.d. from three independent experiments.

8. Figure 3d and 3e have not been mentioned in the text? Where in the manuscript do the authors describe and explain Figure 3d and 3e?

Response:

We appreciate the reviewer's comments. We have corrected this mistake and described the Figure 3d and 3e by inserting these sentences "The above-established QS circuits only have the function of activating transcription. To expand the functionality of the QS system, we tested the potential of using CRISPRi as a switch to achieve signal conversion. Thus, the AHL-induced up-regulation was transduced into a down-regulation function by integrating the CRISPRi system into QS circuits. To confirm whether simultaneously dynamic up-regulation and down-regulation can be achieved in an orthogonal manner, we constructed plasmid *pZE-luxR-P_{QS18}-luxI-eGFP* as the reporter of the up-regulation function. To construct the down-regulation function reporter, the plasmid *plv-P_{QS18}-dCas9-sgRNA* was constructed to repress the *mCherry* expression on the genome of *E. coli* BW25113 (F') (Fig. 3d). As seen from Fig. 3e, in the presence of the AHL and CRISPRi system, the eGFP fluorescence intensity was up-regulated by 7.7-fold, and the

mCherry fluorescence intensity was down-regulated by 82.9% by the end of 24 h. This result demonstrates that coupling the CRISPRi system with the QS variants could generate a dynamic and bi-functional regulation tool for simultaneously up- and down-regulate two sets of genes.” in line 306-318.

9. In addition to titer (mg/L), please provide data in mg/g CDW and mg/L/day. The length of production experiment in this manuscript is perhaps different from the ones in published studies. Cell growth might also affect production titer. Or, better yet, display the results in mg/g of consumed glucose.

Response:

We appreciate the reviewer’s comments. As suggested, we have provided data in mg/g CDW and mg/L/day by modifying the related sentences as “The best-performed producer generated 2.1 g/L SA (546.5 mg/g CDW, 700 mg/L/day)” in line 418-419. As the initial glucose level was only 2.5 g/L in the medium, and we added 20 g/L glycerol as the carbon source for SA production, we believe that it would be not necessary to display the results in consumed glucose. We hope the reviewer can agree with us and we still appreciate the reviewer’s comments.

10. Fig. 4, 5: How about protein abundance? As in many metabolic engineering studies, measurement of proteins is important to understand the difference between static vs dynamic control of gene expression.

Response:

We appreciate the reviewer’s comments. The protein abundance was not measured in this study. Because in this study, we have constructed more than 70 strains for regulation of SA and 4-HC production. In each of these strains, the numbers of regulated enzymes range from 2 to 8. A total of 140 to 350 enzymes need to determine their abundance at the same time, which is time-consuming and labor-cost. In addition, measurement of protein abundance to understand the difference between static vs dynamic control of gene expression is less frequently in some previous studies on dynamic control in metabolic engineering^{9, 10}. Thus, we are not able to provide protein abundance in this manuscript. We hope the reviewer can understand the difficulty and agree with our decision, we still appreciate the reviewer’s comments.

References

1. Savery, N.J. et al. Determinants of the C-terminal domain of the Escherichia coli RNA polymerase α subunit important for transcription at class I cyclic AMP receptor protein-dependent promoters. *J. Bacteriol.* **184**, 2273-2280 (2002).
2. Lawson, C.L. et al. Catabolite activator protein: DNA binding and transcription activation. *Curr. Opin. Struct. Biol.* **14**, 10-20 (2004).
3. De Crombrughe, B., Busby, S. & Buc, H. Cyclic AMP receptor protein: role in transcription activation. *Science* **224**, 831-838 (1984).
4. Berg, O.G. & von Hippel, P.H. Selection of DNA binding sites by regulatory proteins: II. The binding specificity of cyclic AMP receptor protein to recognition sites. *Journal of molecular biology* **200**, 709-723 (1988).
5. Chen, Y. et al. Tuning the dynamic range of bacterial promoters regulated by ligand-inducible transcription factors. *Nat. Commun.* **9**, 1-8 (2018).
6. Wu, J. et al. Developing a pathway-independent and full-autonomous global resource allocation strategy to dynamically switching phenotypic states. *Nat. Commun.* **11**, 1-14 (2020).
7. Lin, Y., Shen, X., Yuan, Q. & Yan, Y. Microbial biosynthesis of the anticoagulant precursor 4-hydroxycoumarin. *Nat. Commun.* **4**, 1-8 (2013).
8. Shen, X. et al. Elevating 4-hydroxycoumarin production through alleviating thioesterase-mediated salicyl-CoA degradation. *Metab. Eng.* **42**, 59-65 (2017).
9. Dinh, C.V. & Prather, K.L. Development of an autonomous and bifunctional quorum-sensing circuit for metabolic flux control in engineered Escherichia coli. *Proc. Natl. Acad. Sci. U.S.A.* **116**, 25562-25568 (2019).
10. Yang, Y. et al. Sensor-regulator and RNAi based bifunctional dynamic control network for engineered microbial synthesis. *Nat. Commun.* **9**, 1-10 (2018).

Reviewers' Comments:

Reviewer #1:

Remarks to the Author:

The authors have adequately addressed my concerns.

Reviewer #2:

Remarks to the Author:

The authors have done a good job addressing the comments, and they have added significant new data as new supplemental figures. I therefore only have two minor comments.

1. In Fig. 5b-the y-axis is listed as relative binding strength, but it is not clear to me how this is determined as the probe with 4-5 CRP binding sites seems to have more unshifted probe than those with 1-3. This would indicate less probe is bound by CRP. I have similar confusion about Fig. S10b.
2. The new Supplemental Figures are nice additions, but they are now out of order with which they appear in the text as they come before Figs. S1-3.

Reviewer #3:

Remarks to the Author:

The manuscript has been modified as our comments, and I think it is ready to be accepted.

Reviewer #4:

Remarks to the Author:

Novelty-related issues have been addressed. However, many additional figures in the rebuttal are only made available to the reviewers and not discussed at all in the main text. Please ensure that the peer review file can be accessed by the readers. Or else, additional experiments conducted by the authors will become meaningless.

Reviewer #1 (Remarks to the Author):

The authors have adequately addressed my concerns.

Reviewer #2 (Remarks to the Author):

The authors have done a good job addressing the comments, and they have added significant new data as new supplemental figures. I therefore only have two minor comments.

1. In Fig. 5b-the y-axis is listed as relative binding strength, but it is not clear to me how this is determined as the probe with 4-5 CRP binding sites seems to have more unshifted probe than those with 1-3. This would indicate less probe is bound by CRP. I have similar confusion about Fig. S10b.

Response:

We thank the reviewer for the further comments. Actually, the bands at the bottom of the picture are unbound DNA probes. The relative binding strength was determined by the ratio of gray values of unbound DNA probe bands in the presence of CRP protein to the gray value of that in the absence of CRP protein. We added same amount of CRP protein and DNA probes into each sample. The DNA probes with high relative binding strength can bind more than one CRP proteins, indicates that less amount of DNA probes is required for protein binding. Thus, the DNA probes with high relative binding strength results in more unbound DNA probes. In the Fig. S5b, the fluorescence of unbound probes with 4-5 CRP binding sites was higher than that of unbound probes with 1-3 CRP binding sites, indicates that the relative binding strength of 4-5 CRP binding sites are stronger. The same to the Fig. S10b.

2. The new Supplemental Figures are nice additions, but they are now out of order with which they appear in the text as they come before Figs. S1-3.

Response:

We appreciate the reviewer's comments. We have carefully re-organized the order of all figures.

Reviewer #3 (Remarks to the Author):

The manuscript has been modified as our comments, and I think it is ready to be accepted.

Reviewer #4 (Remarks to the Author):

Novelty-related issues have been addressed. However, many additional figures in the rebuttal are only made available to the reviewers and not discussed at all in the main text. Please ensure that the peer review file can be accessed by the readers. Or else, additional experiments conducted by the authors will become meaningless.

Response:

We appreciate the reviewer's comments. As suggested, we have included the related figures in the manuscript as main or supplementary display items.

Reviewers' Comments:

Reviewer #2:

Remarks to the Author:

The authors have addressed my concerns.